# Mechanochemical Synthesis of PdO_2_ Nanoparticles Immobilized over Silica Gel for Catalytic Suzuki–Miyaura Cross-Coupling Reactions Leading to the C-3 Modification of 1*H*-Indazole with Phenylboronic Acids

**DOI:** 10.3390/molecules28207190

**Published:** 2023-10-20

**Authors:** Qin Pan, Yong Wu, Aqun Zheng, Xiangdong Wang, Xiaoyong Li, Wanqin Wang, Min Gao, Zainab Bibi, Sidra Chaudhary, Yang Sun

**Affiliations:** 1Department of Applied Chemistry, School of Chemistry, Xi’an Jiaotong University, No. 28 Xianning West Road, Xi’an 710049, China; 2Xi’an Biomass Green Catalysis and Advanced Valorization International Science and Technology Cooperation Base, No. 28 Xianning West Road, Xi’an 710049, China; 3Xixian New District Xingyi Advanced Materials Technology Co., Ltd., Room 1046, 1st Floor, Hongdelou Building No. 20, Science and Technology Innovation Port, Xi’an 712000, China

**Keywords:** ball milling, Suzuki–Miyaura cross-coupling, 1*H*-indazole, PdO_2_ nanoparticle, silica gel

## Abstract

The C-3 modification of 1*H*-indazole has produced active pharmaceuticals for the treatment of cancer and HIV. But, so far, this transformation has seemed less available, due to the lack of efficient C-C bond formation at the less reactive C-3 position. In this work, a series of silica gel-supported PdO_2_ nanoparticles of 25–66 nm size were prepared by ball milling silica gel with divalent palladium precursors, and then employed as catalysts for the Suzuki–Miyaura cross-coupling of 1*H*-indazole derivative with phenylboronic acid. All the synthesized catalysts showed much higher cross-coupling yields than their palladium precursors, and could also be reused three times without losing high activity and selectivity in a toluene/water/ethanol mixed solvent. Although the palladium precursors showed an order of activity of PdCl_2_(dppf, 1,1′-bis(diphenylphosphino)ferrocene) > PdCl_2_(dtbpf, 1,1′-bis(di-*tert*-butylphosphino)ferrocene) > Pd(OAc, acetate)_2_, the synthesized catalysts showed an order of C1 (from Pd(OAc)_2_) > C3 (from PdCl_2_(dtbpf)) > C2 (from PdCl_2_(dppf)), which conformed to the orders of BET (Brunauer–Emmett–Teller) surface areas and acidities of these catalysts. Notably, the most inexpensive Pd(OAc)_2_ can be used as a palladium precursor for the synthesis of the best catalyst through simple ball milling. This work provides a highly active and inexpensive series of catalysts for C-3 modification of 1*H*-indazole, which are significant for the large-scale production of 1*H*-indazole-based pharmaceuticals.

## 1. Introduction

Indazoles refers to a group of bicyclic compounds containing an electron-rich pyrazole and a fused benzene ring [1], which can be regarded as nitrogen-substituted products of indole, but their studies have seemed much less extensive than indole for a long time [1]. There is a ten-π electron aromatic heterocyclic structure on indazoles, naturally giving three tautomers such as 1*H*-, 2*H*-, and 3*H*-indazoles, due to the resonance structures of pyrazole nuclei of indazole (Figure 1a–c) [2]. In practice, 1*H*-indazole was the most thermodynamically stable and abundant tautomer in both gas, aqueous, and metal-containing phases [3]. Meanwhile, 1*H*-indazole derivatives have shown great medical values, including SE063 (HIV protease inhibitor, Figure 1d) [4], Lonidamine (anticancer agent, Figure 1e) [5], and YC-1 (inhibitor of platelet aggregation, Figure 1f) [6], which have aroused wide interests for many years.

Although less stable and less common than the 1*H*-tautomer, 2*H*-indazole plays a key role in the synthesis of many highly useful pharmaceuticals for treatment of diseases like renal cell carcinoma, inflammation, as well as virus infection [3]. On the other hand, probably due to the synthetic inconvenience of 3*H*-indazole or its derivatives, up to date there are few systematic explorations on pharmaceutical applications of 3*H*-indazole [7].

Nowadays, attempts to synthesize and functionalize 1*H*-indazole attract wide and continuous attentions. These approaches can be divided into several types, including nitrosation of indole along with rearrangement [8], Staudinger–aza-Wittig reaction [9], as well as metal-catalyzed N-N bond formation [10]. However, these profiles usually lack substrate and product diversity, obviously depressing the large-scale production of versatile 1*H*-indazole-based pharmaceuticals (Figure 1) [8,9,10].

Therefore, synthetic organic chemists attempted to perform direct functionalization of 1*H*-indazole in order to improve both substrate and product diversities, particularly regarding the functionalization of the C-3 position, mainly due to the key pharmaceutical values of C-3 modified 1*H*-indazoles (Figure 1e,f). However, the C-H bond on the C-3 position of 1*H*-indazole was chemically inert compared to that on 2*H*-indazole [6,11]; metal-free transformation showed little effects [6], so most endeavors were focused on the design and use of metal catalysts for this conversion. For example, Kazzouli and co-workers reported the intermolecular C-H arylation of amino-protected 1*H*-indazole with (hetero)aryl bromide or iodide as a coupling partner, where palladium acetate in association with 1,10-phenanthroline was selected as catalyst, potassium carbonate was selected as a base, and dimethylacetamide was selected as solvent [11]. Yamaguchi and co-workers also employed a catalytic combination of palladium chloride, 1,10-phenanthroline, silver carbonate, and potassium phosphate, along with dimethylacetamide for the direct C–H arylation of 1*H*-indazole with haloarene, eventually leading to a rapid synthesis of YC-1 (an antitumor agent, Figure 1f) [6].

Meanwhile, both Heck [12] and Suzuki–Miyaura cross-couplings [1] have been introduced to this transformation. The Heck reaction often showed poorer efficiency for C-3 functionalization, because the coupling partner demanded a tight (but sometimes irreversible) protection of the N-H group on 1*H*-indazole, otherwise unwanted by-products appeared [12]. In comparison, the suitable coupling partners of Suzuki–Miyaura cross-coupling such as organoboronic acids just needed an easily removable protection of the N-H group, like that from di-*tert*-butyldicarbonate ((Boc)_2_O) [1]. Furthermore, most Suzuki–Miyaura cross-couplings could be accomplished at mild temperature (lower than 80 °C) within a short time (3 h) [13], producing no toxic pollutants [14], deeming them comparable to Heck reactions [12].

In practice, however, there are still some drawbacks on conducting Suzuki–Miyaura cross-coupling reactions, including the high cost of palladium precursors (salts) [13], the lengthy and costly synthesis of palladium catalysts [14], as well as the lack of effective catalyst recovery [13,14]. Therefore, the immobilization of palladium into support was carried out in order to simplify catalyst synthesis and improve catalyst recovery. The support materials covered polymer [15], polysiloxane [16], covalent organic frameworks (COFs) [17], metal-organic frameworks (MOFs) [18], as well as carbon materials [19]. Overall, the structure of a palladium complex immobilized over support usually differs from its homogeneous counterpart, giving lower activity [14]. But, in rare cases, the support may act as an active ligand, reshaping the steric volume, size, and porosity of palladium precursors and showing positive influence on catalytic outputs [16,20].

In addition to traditional catalytic and synthetic protocols, some currently developed approaches like mechanochemistry appear to be a real breakthrough for Suzuki–Miyaura cross-coupling, mainly regarding clean and energy-saving catalytic reactions [21], as well as mechanochemical preparation of catalysts, along with effective recycling [22]. The mechanochemistry means that the chemical transformation, propelled by mechanical energies involving friction, compression, forging, extrusion, and milling as well [23], which is high enough to break, rebuild, or connect chemical bonds, shows opportunities for establishing new reactions or advanced materials [23].

On the one hand, mechanochemistry appears to be a highly attractive technology for achieving a variety of chemical conversions, which may boost brand-new, highly chemo- or stereo-selective, and much safer reactions, because mechanical energies successfully circumvent the drawbacks derived from classical powers [24]. For instance, classical heating such as artificially raising and decreasing temperature would produce unwanted by-products, and heating a closed reactor increases the risk of explosion [23]. Secondly, it was reported that 85% of chemicals applied in the pharmaceutical industry were solvents, but even if recovered, 20–50% of solvents were lost by evaporation [24]. However, mechanochemical reactions need marginal solvents or none at all, which save a huge amount of solvent, leading to much greener and cheaper processes [24].

On the other hand, mechanochemistry contributes to the synthesis or formation of micro- or nano-sized catalysts, nanomaterials, or other submicroscopic materials with great porosity [23]. In comparison, conventional synthetic profiles not only demand lengthy processes, heating, or hazardous and expensive reagents, but also delivered amorphous catalysts (materials) with less activity [23]. Herein, ball milling, as a typical mechanochemical operation, accumulates kinetic, potential, and thermal energies, along with shear and friction forces in a very short time, and then leaves a large variety of defects (changes, dislocations, or polymetallic mixed phases) in the final material, naturally improving the reactivity of synthesized materials [24].

Moreover, ball milling is beneficial to the formation of nanosized chemical structures rather than to degrading them into amorphous ones; the porosity might be increased, and active components would be highly dispersed even on the single-atom scale. For instance, palladium nanoparticles were ever immobilized over single- or multi-walled carbon nanotubes by using palladium acetate as a precursor and carbon nanotubes as support, through ball-milling [25]. This protocol was solventless, rapid, and did not need any reducing reagents or electric current. The nanosized morphology of carbon nanotubes was well maintained where palladium nanoparticles had a size of 1–3 nm [25]. Additionally, the synthesized palladium catalysts accomplished Suzuki–Miyaura cross-coupling within 50 min, giving almost 100% yield, which is much better than classical palladium/carbon catalysts, indicating the palladium nanoparticles were fully dispersed and highly active [25].

Gupton and co-workers prepared a 3D palladium nanoparticle–nickel–graphene–carbon nanotube hybrid catalyst by using ball milling, and synthesized a catalyst which was 10 times more active than commercial palladium/carbon catalysts in Suzuki–Miyaura cross-coupling [26]. More than that, ball milling was also used to construct single-atom palladium components immobilized over MOF (metal–organic framework), and the resulting catalyst showed extremely high activity (turnover frequency of 13,043 h^−1^), selectivity (>99%), and yield (>99%) for Suzuki–Miyaura cross-coupling of bromobenzene with phenylboronic acid [27]. Iron oxide was also employed as support for immobilizing single-atom palladium components through ball milling, and the catalyst showed great activity too [22].

On the other hand, although immobilization of palladium into porous supports may create highly active catalysts, most immobilizations cannot avoid leaching of palladium catalyst into solution, and then inactive palladium black appears [28]. Therefore, ionic liquids (ILs) were always used as liquid support for immobilizing palladium catalyst in order for better recycling [29].

This work aims to construct an efficient, costly, eco-friendly, and recyclable profile for C-3 functionalization of 1*H*-indazole. In practice, various palladium(II) precursors were immobilized into silica gel through ball milling. The catalysts obtained were then employed in the Suzuki–Miyaura cross-coupling of amino-protected 3-iodo-1*H*-indazole with phenylboronic acid. Catalyst recycling was also tested with or without ILs. This work may show values on the manufacture of 1*H*-indazole-based pharmaceuticals.

## 2. Results and Discussion

### 2.1. Elemental Composition and Chemical State on Surface of Catalyst

X-ray photoelectron spectroscopy (XPS) was employed first to detect the elemental composition and chemical state on the surfaces of the synthesized catalysts (depth of 0–3 nm). The synthesis of catalysts is shown in Figure 1. The XPS survey scan is summarized in Figure 2, and the corresponding binding energies and atomic compositions are in Table 1. At first, C0 showed a much lower content of carbon than C1–C3 (Figure 2a vs. Figure 2b–d; C0 vs. C1–C3, C 1s column, Table 1), while C1–C3 showed a considerably lower content of silicon than C0 (C1–C3 vs. C0, Si 2p column, Table 1), and palladium was detected on C1–C3 more than C0 (Figure 2b–d vs. Figure 2a), indicating that silica gel contained marginal carbon species; meanwhile, palladium components were successfully immobilized into silica gel through ball milling and subsequent centrifugation (Figure 1).

Furthermore, the palladium content of C1 seemed much higher than those of C2–C3 (Figure 2b vs. Figure 2c,d; C1 vs. C2–C3, Pd 3d column, Table 1), probably owing to the structural difference in palladium(II) precursors, which may affect the attachment of silica gel with palladium during catalyst synthesis (Figure 1). Additionally, although both C2 and C3 employed phosphorous-containing palladium precursors (Figure 1), there was no phosphorous found on C2 (Figure 2c; C2, Pd 3d column, Table 1), probably indicating that the dppf ligand of the Pd(II) precursor was leached into the solution rather than adsorbed by the silica gel during the synthesis of C2 (Figure 1).

It was important to test the chemical state of the elements on the catalyst surface in order to understand the active component during catalysis. First of all, there were two components which appeared at 342.6 and 337.3 eV on the Pd 3d region of C1 (Figure 3a), corresponding to the 3d_3/2_ and 3d_5/2_ photoelectrons of Pd^4+^, respectively [30,31]. For the purpose of comparison, it was previously reported that Pd^0^ showed the peaks of 3d_3/2_ and 3d_5/2_ photoelectrons at 341.3–340.9 and 335.8–335.4 eV [30,31]. The Pd 3d regions of C2 and C3 showed very similar peak contours along with binding energies compared to C1 (Figure 3b,c vs. Figure 3a). Therefore, the palladium species formed on C1–C3 should have tetravalence.

In order to further understand the composition of the palladium species on C1–C3, wide-angle (2*θ* = 10°–80°) X-ray diffraction (XRD) was performed as shown in Figure 4. A series of diffractions occurred at 2*θ* = 35.15°, 58.15°, and 67.15° (grey cubes, Figure 4b), probably corresponding to 101, 220, and 112 indices derived from PdO_2_ (palladium oxide, PDF No. 34-1101). The diffractions of PdO_2_ may also be found in the wide-angle XRD of C2–C3 (grey cubes, Figure 4c,d vs. Figure 4b). Therefore, it seemed that PdO_2_ probably emerged as the metal-containing phase after ball milling, no matter which kind of divalent palladium precursor was used (Figure 1). Herein, Pd^2+^ coming from various divalent palladium precursors was hydrolyzed into Pd(OH)_2_ under ball milling, which was further oxidized and dehydrated into PdO_2_ under drying (Figure 1).

Next, the detection of carbon species on the synthesized catalysts shows additional information on immobilization. In fact, C1–C3 showed three very similar components on their C 1s regions, including the first one which appeared at 284.6–284.7 eV, the second at 285.7–285.9 eV, as well as the last at 288.4–288.6 eV (Figure 5b–d), which can be ascribed to carbons stemming from the saturated hydrocarbon (sp^3^ hybridization), C–O bond, and carboxyl group, respectively [32]. These components may be derived from the organic ligands of the palladium precursors (Figure 1). In comparison, however, C0 only showed two components that appeared at 285.4 and 287.2 eV, corresponding to a C–O bond and carboxyl group, respectively [32], actually representing the carbon residues of silica gel (C0, Figure 1).

Then, this aroused further interest to test other elements like silicon on the catalyst surface. In general, the binding energies of Si 2p photoelectrons can be found at 103.2–104.1 eV for C0–C3 (Appendix A), which could be attributed to Si^4+^ of the SiO_2_ phase [33]. Accordingly, C0–C3 all showed a broad XRD band in the range of 2*θ* = 15°–35°, centered at 23° (Figure 4a), corresponding to the typical diffraction of the silicate backbone coming from C0 (Figure 1) [34].

C2 showed a similar XRD contour to C1 (Figure 4c vs. Figure 4b), but the diffraction of grunerite might be detected (dark cube, Figure 4c; PDF No. 44-1401, Fe_7_Si_8_O_22_(OH)_2_), probably indicating that the dppf ligand was decomposed and transformed into Fe-Si mixed oxide during ball milling (Figure 1). C3 showed a higher Si 2p photoelectron binding energy than C0–C2 (Appendix A vs. Appendix A), indicating that some new Si-containing phases appeared during immobilization (Figure 1). On one hand, in addition to PdO_2_, the 002 diffraction of SiP may appear on the XRD of C3 (hollow cube, Figure 4d; PDF No. 29-1133, SiP). On the other hand, two peaks occurred at 129.2 and 122.8 eV on the P 2p region of C3 (Appendix A), probably corresponding to the 2p_1/2_ and 2p_3/2_ photoelectrons of P^4−^ [35], which were both lower than those of phosphorous with high valence coming from P–O or P=O bonds [36].

### 2.2. Textural and Other Properties of Synthesized Catalyst

In order to further understand the difference in catalysts, the textural and other physicochemical properties of the synthesized samples were tested. At first, all the catalysts (C0–C3) showed typical type IV isotherms along with H3 type hysteresis loops (Figure 6a–d), clearly indicating they had mesoporous structures [37], which is further established by the pore size distributions (Figure 6a’–d’) and high BET surface areas (Table 2). This result meant that ball milling would not destroy the mesoporous structure of silica gel during the immobilization of palladium (Figure 1).

C1–C3 showed a lower BET surface area, total pore volume, micropore volume, as well as higher bulk density than C0 (Table 2), indicating palladium was incorporated into both the mesopores and micropores of C0 during immobilization (Figure 1). Furthermore, C1–C3 showed an order of BET surface areas of C1 > C3 > C2 (Table 2), proposing that the ligand of the palladium precursor played a key role in determining the adsorption amount of palladium over support (Figure 1). Meanwhile, the order of acid amount was detected as C1 > C3 > C2 > C0 (Table 2), proposing that C0 had Lewis or Brönsted acidities by nature, but the immobilization of palladium would further increase the acidity, mainly because the palladium atom may provide more empty orbitals than silicon, and various ligands of palladium precursors showed different influences on product acidity (Figure 1).

### 2.3. Functional Group and Thermal Stability of Synthesized Catalyst

FT-IR spectroscopy was carried out to detect the functional group of the synthesized catalysts. As shown in Appendix A), all the tested samples (C0–C3) showed a broad band centered at 3450 cm^−1^, representing the O-H stretching vibration of hydroxyl groups on the surface of silica gel (Appendix A). Furthermore, the peaks that appeared at 1622 cm^−1^ on C0–C3 could be ascribed to the O-H bending vibration of hydroxyl groups (Appendix A) [40].

Then, there were two following peaks which occurred at 1068 and 798 cm^−1^ (Appendix A), corresponding to unsymmetric and symmetric Si–O stretching vibrations, respectively [41]. Moreover, the contours around 1068 cm^−1^ for C0–C2 were much broader than that for C3 (Appendix A vs. Appendix A), probably indicating the backbone of C3 had been changed to a large extent, also in association with the XRD of C0–C2 vs. C3 (Figure 4d vs. Figure 4a–c). Additionally, both C3 and C2 showed the stretching vibration of the Pd-O bond at 492 cm^−1^ (Appendix A), which was red-shifted slightly on C1 (Appendix A), probably indicating that both dppf and dtbpf ligands had residues after ball milling, which affected the coordination environments of Pd^4+^ (Figure 1).

UV-Vis spectroscopy was employed to study the immobilization from another point of view. Firstly, all the tested samples (C0–C3) showed adsorption at 284 nm (Appendix A), corresponding to the charge transfer transition on the organic species of samples [42]. C0–C2 showed no adsorptions above 325 nm (Appendix A), but there were two bands centered at 362 and 450 nm on the UV-Vis spectrum of C3 (Appendix A), corresponding to the ligand to metal charge transfer (LMCT) transition of the metal species [42] and the charge transfer transition of metals [42], respectively. This phenomenon indicated again that C3 had a quite different backbone than C0–C2, where a new Si-containing phase may appear, as shown in the XRD (Figure 4d) and FT-IR (Appendix A).

It was also interesting to study the composition of the synthesized catalyst by using TGA. C0 showed a sharp weight loss of 5.62% at 35–150 °C (black line, Figure 7), proposing the release of adsorbed water. The following weight loss of 5.46% that occurred at 150–600 °C tended to be relatively flat (black line, Figure 7), indicating the decomposition of organic residues on C0.

After the immobilization of Pd(OAc)_2_ into silica gel (Figure 1), the resulting C1 showed the first weight loss of 3.82% at 35–150 °C, along with the following one of 4.51% at 150–600 °C (red line, Figure 7). In comparison, C0 showed a higher total weight loss (11.08%) than C1 (8.33%) at 30–600 °C (black vs. red, Figure 7), obviously indicating that ball milling removed the adsorbed water and organic residues in silica gel, facilitating the immobilization of palladium (C1 vs. C0, Figure 1).

Furthermore, both C2 and C3 showed much lower total weight losses than C1 (green and blue vs. red, Figure 7), probably owing to their lower palladium contents compared to C1 (Table 1), reflecting the different effects of the palladium precursors used in the synthesis of C1–C3 (Figure 1). Moreover, the TGA curve of C3 coincided with C2 at 30–200 °C but exhibited a much higher weight loss at 200–600 °C (blue vs. green, Figure 7), proposing that C3 contained the same amount of adsorbed water or volatile organic species as C2, while the less volatile components on C3 seemed more abundant than those on C2 (Figure 1).

### 2.4. Morphology and Internal Structure of Synthesized Catalyst

C0 was commercial silica gel with size 300 mesh (about 8.47 μm on average). According to the SEM observation, C0 had a comparatively smooth and flat surface with a few wrinkles (Figure 8a,a’), reflecting the original morphology of silica gel (raw material, Figure 1).

C1 turned out to be a mass of irregular particles of a size of 1–5 μm (Figure 8b,b’), obviously indicating that ball milling had at least halved the size of silica gel (C0) (Figure 1). In association with the TEM observation, it can be seen that the much denser particles featuring a size of 10–20 nm could be ascribed to PdO_2_ (Figure 9a, Figure 1).

C2 was composed of particles of a size of 1–3 μm, along with their agglomerates (Figure 8c,c’). Simultaneously, C2 showed a similar internal structure compared to C1, including the sizes of both blocks and mesopores (Figure 9b vs. Figure 9a), which also conformed to those found through the N_2_ adsorption–desorption experiments (Table 2), probably suggesting the immobilization of PdCl_2_(dppf) would not degrade the backbone of silica gel as that of Pd(OAc)_2_ (Figure 1).

Although C3 showed a similar morphology to C2 according to SEM (Figure 8d,d’ vs. Figure 8c,c’), C3 exhibited a much looser internal structure than C2, particularly regarding the much larger pore size of C3 (Figure 9c vs. Figure 9b). Accordingly, it can be seen that C3 exhibited higher BET surface area and total pore volume, as well as lower bulk density than C2 (Table 2). Therefore, the ligand of the palladium precursor played an important role in controlling the product structure (Figure 1).

With the data obtained so far, it seemed important to compare the sizes of both bulk and PdO_2_ particles derived from different sources. On one hand, C0 showed a bulk particle size coming from BET surface area at 43 nm (C0, *d*_S_, Table 2), while that from SEM turned out to be 40 μm (C0, *d*_SEM_, Table 2; Figure 8a,a’), indicating that the particles of a size of 43 nm were the smallest units of N_2_ adsorption and desorption, whose agglomeration actually constituted the much larger particles of a size of 40 μm. The same trends were also found in C1–C3 (*d*_S_ and *d*_SEM_, Table 2).

On the other hand, it was found that C1 had PdO_2_ particles featuring a size of 10–20 nm under TEM detection (dark dots, Figure 9a), and the calculated size was 25 nm on the basis of XRD (C1, *d*_PdO2_, Table 2), whose similarity approved the presence of PdO_2_ nanoparticles on C1 (Figure 1). Furthermore, both C2 and C3 showed much larger PdO_2_ nanoparticle sizes than C1 according to XRD (*d*_PdO2_, Table 2); meanwhile, the PdO_2_ nanoparticles may have been dispersed into the denser blocks on their TEM images (Figure 9b,c).

### 2.5. Catalytic Suzuki–Miyaura Cross-Coupling Reaction

#### 2.5.1. Preparation of Iodinated 1H-indazole Substrate

In order to carry out the C-3 modification of 1*H*-indazole (compound **1**) through catalytic Suzuki–Miyaura reaction, substrate **1** needed to be first iodinated almost stoichiometrically according to a previous method [43], whose amino group could be subsequently protected by di-*tert*-butyldicarbonate ((Boc)_2_O) [43]. Herein, iodination can be conducted without affecting the amino group, but the amino group had to be protected before the cross-coupling, otherwise N-substituted products may appear [1,12]. After the reaction, the Boc group can be removed in acid solution [43].

#### 2.5.2. Reactions Using Two Substrates

According to Table 3 and Table 4, the averaged conversion of entries 3–10 in Table 3 (60%) was lower than that of same entries in Table 4 (62%). As for entries 11–15 in Table 3 and Table 4, Table 3 (64%) still gave fewer outputs than Table 4 (69%). Additionally, the averaged conversion of entries 16–19 in Table 3 (66%) was also lower than that in Table 4 (72%). Therefore, 3-methoxycarbonylphenylboronic acid (compound **2**, Table 3) seemed to be a less effective nucleophile than its para isomer (compound **4**, Table 4) under various catalytic profiles. It was previously confirmed that methoxycarbonyl is an electron withdrawing group (EWG) [44]. When methoxycarbonyl was *ortho-*, *meta-,* or *para*-oriented on phenylboronic acid, the homolytic bond dissociation enthalpy (BDE) of the C–B bond showed an order of *ortho-* < *para-* < *meta*-oriented, which was also the order of the C–B bond strength [44], meaning the cleavage of the C–B bond of compound **4** needed less energy than that of compound **3**.

#### 2.5.3. Screening of Catalysts

Most Suzuki–Miyaura cross-coupling reactions were accomplished within 8 h, having no detectable by-products. For the coupling of *tert*-butyl-3-iodo-1*H*-indazole-1-carboxylate (compound **1**) with **2**, C0 (silica gel) did not deliver any cross-coupled products (entries 1–2, Table 3 and Table 4), clearly indicating that the silicate backbone showed no catalytic activity at all.

It is noteworthy that C1 showed much higher conversion and yield than Pd(OAc)_2_ (palladium precursor, Figure 1; entries 4 vs. 3, Table 3). Furthermore, C2 and C3 showed higher outputs than their palladium precursors, such as PdCl_2_(dppf) and PdCl_2_(dtbpf), respectively (entries 12 vs. 11, 17 vs. 16, Table 3). In the meantime, when compound **2** was replaced with **4** as a new nucleophile, the same trends were found (entries 4 vs. 3, 12 vs. 11, 17 vs. 16, Table 4).

Therefore, the PdO_2_ nanoparticles (size of 25 nm, C1, Table 2; Figure 9a) immobilized over silica gel under ball milling had much higher catalytic activity than Pd(OAc)_2_; meanwhile, the PdO_2_ nanoparticles (size of 38–66 nm, C2–C3, Table 2; Figure 9b,c) of C1–C2 also performed better than their palladium precursors such as PdCl_2_(dppf) and PdCl_2_(dtbpf). Obviously, ball milling converted palladium(II) precursors into PdO_2_ nanoparticles, leading to higher activity.

On the other hand, the palladium precursors showed an order of activity of PdCl_2_(dppf) > PdCl_2_(dtbpf) > Pd(OAc)_2_ for transformations of both compounds **2** (entries 11 vs. 16 vs. 3, Table 3) and **4** (entries 11 vs. 16 vs. 3, Table 4), clearly indicating that the ligand of the palladium precursor showed key influences on the homogeneous catalytic efficiency. There were three steps in the Suzuki–Miyaura cross-coupling, including oxidative addition, transmetallation, and reductive elimination [13]. The ligand of the palladium precursor played a key role in oxidative addition, directly affecting the catalytic outputs [13].

However, the activity order of the synthesized catalysts turned out to be C1 > C3 > C2 for the transformation of not only compounds **2** but also **4** (entries 4 vs. 17 vs. 2 in Table 3 and Table 4). At first, the synthesized catalysts showed an order of BET surface areas of C1 > C3 > C2, while their bulk densities gave the opposite trend of C1 < C3 < C2 (Table 2). The porosity of the synthesized catalysts was directly controlled by the ligand of the palladium precursor that was used in ball milling (Figure 1), which greatly affected the dispersion of the substrate into the catalytic center.

Secondly, the acid amounts of the synthesized catalysts exhibited an order of C1 > C3 > C2 (Table 2), indicating that ball milling of silica gel with Pd(OAc)_2_ prepared the catalysts featuring the highest Lewis and Brönsted acidities; that those with PdCl_2_(dtbpf) did less so; while those with PdCl_2_(dppf) exhibited the lowest (Figure 1; entries 4 vs. 17 vs. 2 in Table 3 and Table 4). In synthesis, it was proposed that acetate anion (OAc^−^) accelerated the hydrolysis of Pd^2+^ into hydroxide, which was subsequently dehydrated to PdO_2_ nanoparticle under ball milling. But in comparison, both dppf and dtbpf could not accelerate this process to the same extent (Figure 1).

Furthermore, although the structures of dppf and dtbpf looked similar (Figure 1), PdCl_2_(dppf) seemed better than PdCl_2_(dtbpf) in the catalytic conversion of compounds **2** and **4** (entries 11 vs. 16, Table 3 and Table 4), meaning that dppf was able to stabilize the palladium intermediate [PhPdXLn] better than dtbpf in oxidative addition [13], probably because the phenyl group of dppf was much smaller than the *tert*-butyl of dtbpf, leaving less steric hindrance in the formation of [PhPdXLn] during oxidative addition [13].

But in heterogeneous catalysis, C3 coming from PdCl_2_(dtbpf) became more powerful than C2 from PdCl_2_(dppf) (entries 17 vs. 12, Table 3 and Table 4). Therefore, ball milling actually changed the original palladium precursors into other catalytic components like PdO_2_ nanoparticles (Figure 1). In practice, dtbpf performed better than dppf during the oxidative hydrolysis of Pd^2+^ and in dehydration under ball milling, which is supported by the higher BET surface area, total pore volume, and acid amount of C3 compared to those of C2 (Table 2). Additionally, C3 showed a different XRD pattern than C2 (Figure 4d vs. Figure 4c), where the phase of SiP may have appeared (Figure 4d and Appendix A), which may have decorated surface of the PdO_2_ nanoparticle, leading to better catalytic outputs.

From another point of view, the yield stemming from the protocol of Table 4 catalyzed by C2 (entry 12, Table 4; yield of 88%) was much higher than that from the same reaction catalyzed by PdCl_2_(dppf) (yield of 65%) [43]. Furthermore, C2 also showed a very close yield compared to the catalytic combination of PdCl_2_(dppf) with BMImPF_6_ [45]. These results clearly indicate that the ball milling of the divalent palladium(II) precursor with silica gel produces a highly active palladium catalyst.

#### 2.5.4. Effects of Loading Amounts of Catalyst and Base

When the loading amount of C1 was halved from 4 mol% to 2 mol%, both the conversion and yield for the transformation of compound **2** were increased by 2% (entries 5 vs. 4, Table 3). If compound **2** was replaced with **4**, the catalytic outputs including the conversion and yield were slightly decreased (entries 5 vs. 4, Table 4). Overall, the use of C1 (the heterogeneous catalyst with highest activity) was not very sensitive to the loading amount of the catalyst.

Herein, when C1 was reduced by half in the transformation of compound **2** (meta isomer), the agglomeration of C1 was weakened, and the dispersion of **2** into the internal pores of C1 became easier, leading to better outputs. If compound **2** was replaced with **4** (para isomer), the catalytic outputs were slightly decreased when the catalyst loading was halved, suggesting that the cross-coupling of **4** may occur on the surface of C1 more than in its internal pores, simultaneously indicating the textural structure of C1 preferred the meta isomer (**2**, smaller) to the para one (**4**, larger), mainly owing to the size of the substrate.

The loading amount of the base (Na_2_CO_3_) also affected the catalytic outputs significantly. In practice, when the loading of Na_2_CO_3_ was decreased by half, the catalytic outputs including the conversion and yield were degraded remarkably, not only for the transformation of the meta isomer (**2**, entries 6 vs. 4, Table 3), but also for that of the para isomer (**4**, entries 6 vs. 4, Table 4). According to the proposed mechanism [13], Na_2_CO_3_ was dissolved in water and released OH^−^, and the concentration of OH^−^ affected the appearance of the [PhPdOHLn] intermediate in transmetallation, involving the forming rate and concentration. Clearly, a high concentration of OH^−^ in the catalytic solution contributed to the catalytic outputs.

#### 2.5.5. Effect of Temperature

When the temperature was decreased gradually, the cross-coupling of compound **2** catalyzed by C1 decreased the conversions and yields sharply (entries 4 vs. 9 vs. 10, Table 3). The same tendency was also observed in the cross-coupling of compound **4** (entries 4 vs. 9 vs. 10, Table 4). Although much of the literature has reported that catalytic Suzuki–Miyaura cross-coupling reactions can be carried out at mild temperatures [13], C1 still needed 80 °C to guarantee high catalytic outputs, no matter which kind of phenyl substrate was used, supposing a higher temperature would facilitate the formation of an active intermediate during catalysis.

#### 2.5.6. Effect of Ionic Liquid

In order to utilize the many great properties of IL, two imidazolium ILs such as BMImX (BMIm^+^ = 1-*n*-butyl-3-methylimidazolium, X^−^ = BF_4_^−^, PF_6_^−^) were employed as co-solvents. However, the combination of BMImBF_4_ with THE (toluene/H_2_O/EtOH, Table 3 and Table 4) did not show any positive effects on the transformation of both compounds **2** and **4** (entry 2 for Table 3 and Table 4), indicating that the BMIm^+^ ions were completely inactive in catalyzing the present cross-coupling.

Next, when C1 was selected as a catalyst, the introduction of either BMImBF_4_ or BMImPF_6_ did not improve the catalytic outputs for the conversions of both compounds **2** and **4**, and even depressed the reactions (entries 7–8 vs. 4, Table 3 and Table 4). Moreover, neither BMImBF_4_ nor BMImPF_6_ showed positive effects on the reactions of compounds **2** and **4** catalyzed by C2 (entries 14–15 vs. 12, Table 3 and Table 4) and C3 (entries 18–19 vs. 17, Table 3 and Table 4).

It has been widely reported that the immobilization of palladium species into ionic liquid can provide more powerful catalysts for Suzuki–Miyaura cross-coupling, mainly because this kind of immobilization can increase the acidity of the palladium component, stabilize the active intermediate, inhibit the formation of non-catalytic palladium black, as well as improve catalyst recycling [45,46]. However, the suitable palladium components may be limited to homogeneous palladium salts or complexes rather than supported palladium materials. Herein, the catalytically active components of C1–C3 were PdO_2_ nanoparticles immobilized over silica gel, whose surface and internal pores may have been largely blocked by ILs, subsequently leading to very poor dispersion of the substrate into catalytic centers.

Furthermore, BMImPF_6_ seemed better than BMImBF_4_ in the C1 and C3 catalyzed reactions for two substrates (entries 8 vs. 7, 19 vs. 18, Table 3 and Table 4), but C2-facilitated transformation showed a contrary tendency (entries 15 vs. 14, Table 3 and Table 4). Herein, BF_4_^−^ was water-soluble and PF_6_^−^ was immiscible with water [46]. Therefore, it seemed that the surfaces and internal pores of C1 and C3 were lipophilic, while C2 was hydrophilic, which affected their affinity to different ILs, leading to different outputs.

#### 2.5.7. Effect of Catalyst Recycling

The results of catalyst recycling are summarized in Figure 10 and Figure 11. At first, the recycling of synthesized catalysts including C1–C3 themselves showed slightly decreased or even improved yields in the cross-couplings of **1** with **2** (green columns, Figure 10b–d) and **4** (green columns, Figure 11b–d) during three instances of use. However, use of ILs as recycling media such as BMImBF_4_ and BMImPF_6_ gave sharply degraded or continuously low outputs for the cross-couplings of **1** with both **2** (red and blue columns, Figure 10b–d) and **4** (red and blue columns, Figure 11b–d).

Therefore, the PdO_2_ nanoparticles fixed over silica gel not only showed great activity in the fresh round, but also exhibited considerable endurability during their continuous use. However, the introduction of imidazolium ILs as recyclable media was not effective; it blocked the dispersion of substrates into PdO_2_ nanoparticles, and also caused the decomposition of catalytic centers during continuous catalytic rounds.

### 2.6. Proposed Mechanism for Heterogeneous Suzuki–Miyaura Cross-Coupling

A proposed reaction process is summarized in Figure 12. The PdO_2_ nanoparticles fixed on the surface of C1 were first activated and reduced to Pd(0) species by the internal electron transfer of C1, leading to catalytically active TS1 [13]. Next, the organic iodide (electrophile, compound **1**) was coordinated to TS1, meanwhile the surficial PdO_2_ nanoparticles were oxidized to Pd(II) species under the present conditions, like oxidation of O_2_ in air [13].

The resulting TS2 was further coordinated by the substituted phenyl anion (TS2–TS3, Figure 12). Herein, the hydrolysis of the Na_2_CO_3_ (base) provided OH^−^, whose nucleophilic attack to the boron in compound **4** gave a phenyl anion. Furthermore, TS3 underwent reductive elimination, yielding not only the target product (compound **5**), but also TS1 for recycling (TS3–TS1, Figure 12) [13].

In comparison, the ligand of the palladium(II) precursor stabilized the intermediate during oxidative addition, but this effect became marginal on the surface of the PdO_2_ nanoparticles, where surficial hydroxyl groups turned out to be great substitutes for those ligands. On the other hand, although the immobilization of homogeneous palladium(II) complexes into ILs enhanced both oxidative addition and transmetallation [45,46], the dispersion of PdO_2_ nanoparticles immobilized over silica gel into ILs was not successful, probably because the surface of the PdO_2_ nanoparticles was saturated and the internal pores were blocked by ILs.

## 3. Experimental Section

### 3.1. Starting Materials

The silica gel (SiO_2_, purity of 99%, 300 mesh, for both immobilization and column chromatography) was purchased from Qingdao Haiyang Chemical Co., Ltd. (Qingdao, China). The palladium(II) acetate (Pd(OAc)_2_, purity of 99%, Pd content of 47%), 1,1′-bis(diphenylphosphino)ferrocene-palladium(II)dichloride dichloromethane complex (PdCl_2_(dppf), purity of 98%, content of Pd > 13%), [1,1′-bis(di-*tert*-butylphosphino)ferrocene]dichloropalladium(II) (PdCl_2_(dtbpf), purity of 98%), 1*H*-indazole (purity of 98%), di-*tert*-butyldicarbonate ((Boc)_2_O, purity of 99%), 4-dimethylaminopyridine (DMAP, purity of 99%), 3-methoxycarbonylphenylboronic acid (compound **2**, purity of 97%), and 4-methoxycarbonylphenylboronic acid (compound **4**, purity of 97%) were all bought from Sigma-Aldrich (St. Louis, MO, USA).

Two imidazolium ILs, abbreviated as BMImX (BMIm^+^ = 1-*n*-butyl-3-methylimidazolium; X^−^ = BF_4_^−^, PF_6_^−^; both purities of 98%) were commercially available from Alfa Aesar (Thermo Fisher Scientific (China) Co., Ltd., Shanghai, China). Deuterium reagents such as CDCl_3_ and CD_3_OD were bought from Shanghai Macklin Biochemical Technology Co., Ltd. (Shanghai, China). Distilled water was prepared in our laboratory. Other organic solvents were provided by local distributors and used after purification in our laboratory.

### 3.2. Synthetic and Analytical Instruments

Ball milling was carried out on a YJKS-Speediness Grind Machine, Foshan Tenghao Instrument Technology Co., Ltd. (Foshan, China), featuring two-vessel specification, a voltage of 220 V, as well as 370 W power. The milling beads of ZrO_2_ had a size of 18 mm. X-ray photoelectron spectroscopy (XPS) was performed on a Kratos Axis Ultra DLD (Kratos Co., Ltd., Manchester, UK), whose lighting source selected monochromatic Al K α X-rays (1486.6 eV). During data processing, the binding energy scale was first calibrated by setting the C 1s peak at 284.8 eV. The peaks were then fitted by using the Gaussian–Lorentz (G/L) product function with 30% Lorentzian component.

The wide-angle (2*θ* = 10–80°) X-ray diffractions were performed on a Philips X’Pert Pro diffractometer (PANalytical B.V. Co., Ltd., Almelo, Holland), whose X-ray source was Cu-Kα radiation (λ = 1.5418 Å), along with a diffraction angle interval of 0.05° s^−1^. The porosity parameters including BET surface area, total pore volume, micropore volume, and pore size were determined on Micromeritics ASAP 2020 (Micromeritics Instruments Corporation, Norcross, Atlanta, GA, USA). The nitrogen adsorption isotherms were measured at 77.35 K, and the samples were degassed in vacuum at 150 °C before testing. The surface area was calculated by using the multi-point Brunauer–Emmett–Teller (BET) method in light of the adsorption data with a relative pressure P/P_0_ at 0.06–0.3. Total pore volume was derived from the nitrogen adsorbed at P/P_0_ = 0.97. Both pore volume and pore size were calculated by using the Barrett–Joyner–Halenda (BJH) method.

The acid amounts including both Lewis and Brönsted acidities of synthesized samples were determined by *n*-butylamine titration in association with coeruleum bromocresolis as indicator. This protocol was described previously [47]. The sample (300 mg) was first combined with *n*-butylamine solution (25.00 mL, 0.05 mol L^−1^ in toluene) into a conical flask (250 mL). After shaking for 5 min under cover, 2-proponal (100 mL) and coeruleum bromocresolis (one drop, diluted solution) were consequently added. The solution obtained was then titrated with HCl solution (0.025 mol L^−1^ in water), and the end point of titration was judged as when the solution color changed from blue to yellow. The acid amount of samples was evaluated as the amount of *n*-butylamine adsorbed, through the subtraction of the residues (*n*-butylamine) in the solution (determined by HCl titration) from the pre-added total *n*-butylamine.

FT-IR spectra were obtained on a Bruker Tensor 27 spectrometer when the samples were dispersed into potassium bromide pellets. UV–Vis spectroscopy was obtained on UV 1800, Shimadzu. Thermogravimetric analysis (TGA) was performed on NETZSH TG 209C, having TASC 414/4 controller, and samples were heated under nitrogen protection (N_2_ flow, 40 mL min^−1^), along with a heating rate of 10 °C min^−1^ at 30–600 °C. Scanning electron microscopy (SEM) was performed on JEOL JSM-6700F (JEOL Ltd., Tokyo, Japan) at 20.0 kV in the absence of Au coating. Transmission electron microscopy (TEM) was conducted on JEOL JEM-200CX (JEOL Ltd., Tokyo, Japan) at 120 kV.

As for the characterizations of catalytic products, the melting points were tested on WRR-Y melting point apparatus (Shanghai INESA Physico-Optical Instrument Co., Ltd., Shanghai, China). ^1^H NMR was detected on a Bruker ADVANCE III instrument (using 400 MHz practically), and ^13^C NMR also on Bruker ADVANCE III (using 101 MHz practically). The C, H, and N elemental analyses were carried out on an Elementar VarioEL III instrument (Langenselbold, Germany).

### 3.3. Synthesis of Catalysts

As shown in Figure 1, Pd(II) precursor (3 g, including Pd(OAc)_2_, PdCl_2_(dppf) and PdCl_2_(dtbpf)) and silica gel (30 g, C0) were combined with distilled water (100 mL) into one vessel on a grind machine, then 20 beads of ZrO_2_ were added. After ball milling for 30 min, the resulting mixture was collected and centrifuged. The liquid phase was removed, and the left solids were transferred to a bake oven and dried at 90 °C for 6 h, yielding C1 (31.9 g), C2 (31.7 g), C3 (32.1 g).

### 3.4. Synthesis of Tert-butyl-3-iodo-1H-indazole-1-carboxylate (Compound **1**)

The 3-iodo-1*H*-indazole, as a synthetic precursor of compound **1**, was prepared through direct iodination of 1*H*-indazole according to the literature [48]. I_2_ (32.0 g, 0.128 mol) and KOH pellets (13.44 g, 0.24 mol) were combined into a DMF solution (120 mL) of 1*H*-indazole (7.54 g, 0.064 mol) at 25 °C. After magnetic stirring (300 rpm) for 1 h; the reaction mixture was poured into aqueous NaHSO_3_ (10%, 300 mL) and then extracted with diethyl ether (2 × 150 mL). The organic layers were combined and washed with distilled water (2 × 150 mL) and brine (2 × 150 mL), dried over anhydrous MgSO_4_, and the organic solvent was evaporated under reduced pressure to produce a lightly yellow solid (14.57 g, yield of 97%). M.p. 142.0–143.0 °C (lit. m.p. 141 °C [1]). ^1^H NMR (400 MHz, CD_3_OD) δ_H_, ppm: 7.19 (1H, m, Ar*H*), 7.42 (1H, t, *J* = 4.0 Hz, Ar*H*), 7.44 (1H, t, *J* = 4.0 Hz, Ar*H*), 7.50 (1H, m, Ar*H*) (Appendix A). ^13^C NMR (101 MHz, CD_3_OD) δ_C_, ppm: 140.75, 127.55, 127.19, 121.32, 120.60, 110.07, 92.02 (Appendix A). Analysis calcd for C_7_H_5_N_2_I: C, 34.42; H, 2.04; N, 11.47. Found: C, 34.64; H, 1.94; N, 11.39.

The 3-iodo-1*H*-indazole (2.44 g, 10 mmol), (Boc)_2_O (2.4 g, 11 mmol), DMAP (60 mg, 0.48 mmol), and triethylamine (2.2 mL, 1.52 g, 15 mmol) were mixed with CH_3_CN (20 mL) into a round-bottom flask (250 mL) having a condenser sealed with balloon. After vigorous stirring at 25 °C for 10 h, the dark orange solution was evaporated to dryness by using rotary evaporation. The residue was then redissolved into diethyl ether (150 mL), and then the organic layer was further washed with brine (2 × 50 mL), dried over anhydrous MgSO_4_, and then concentrated through rotary evaporation to yield crude product. The crude product was purified by column chromatography (SiO_2_, 300 mesh; petroleum ether/ethyl acetate, 5/1, *v*/*v*; a few drops of triethylamine added in eluent) to produce compound **1** (yellow solid, 3.08 g, 90% yield). M.p. 118.0–119.5 °C. ^1^H NMR (400 MHz, CDCl_3_) δ_H_, ppm: 1.63–1.69 (9H, m, *tert*-butyl), 7.27 (1H, m, Ar*H*), 7.39 (1H, d, *J* = 8.0 Hz, Ar*H*), 7.49 (1H, t, *J* = 8.0 Hz, Ar*H*), 8.01 (1H, d, *J* = 8.0 Hz, Ar*H*) (Appendix A). ^13^C NMR (101 MHz, CDCl_3_) δ_C_, ppm: 148.44, 139.67, 130.26, 130.07, 124.29, 122.07, 114.64, 103.05, 85.59, 28.22. Analysis calcd for C_12_H_13_N_2_O_2_I: C, 41.86; H, 3.77; N, 8.13. Found: C, 41.90; H, 3.65; N, 8.52.

### 3.5. Catalytic Suzuki–Miyaura Cross-Coupling Reaction

#### 3.5.1. Non-Ionic Liquid-Facilitated Reaction and Catalyst Recycling

As shown in Figure 10 and Figure 11, compound **1** (2.0 mmol, electrophile), compound **2** and **4** (2.0 mmol, nucleophile), Pd catalyst (C1–C3, Pd of 2–4 mol%, based on compound **1**; C0, equal Si to C1; according to XPS, Table 1), and anhydrous Na_2_CO_3_ (2.0–4.0 mmol) were combined with the THE solvent system (toluene, 8 mL; H_2_O, 4 mL; EtOH, 2 mL) into a round-bottom flask (100 mL) with a condenser sealed with a balloon. Then, the resulting mixture was magnetically stirred at 80 °C for 8 h.

After cooling to room temperature, the mixture was filtrated under reduced pressure, and the residues (catalyst) were collected, dried in air, and then recycled after consumables (substrates, solvent, base) were added. The filtrate was extracted with diethyl ether (3 × 50 mL); the organic layers obtained were washed with saturated NaHCO_3_ solution (50 mL), distilled H_2_O (50 mL), and brine (50 mL). After being dried over anhydrous Na_2_SO_4_ and filtered, the solution was concentrated under reduced pressure and the residue was purified by column chromatography (SiO_2_, 300 mesh; petroleum ether/ethyl acetate, 4/1, *v*/*v*; adding ten drops of triethylamine in eluent of 200 mL). The cross-coupling products (compound **3** or **5**) were obtained as orange and white solids, respectively.

#### 3.5.2. Ionic Liquid-Facilitated Reaction and Catalyst Recycling

As shown in Figure 10 and Figure 11, compound **1** (2.0 mmol, electrophile), and Pd catalyst (C1–C3, Pd of 2–4 mol%, based on compound **1**; C0, equal Si to C1; according to XPS, Table 1) were combined and fully dispersed into imidazolium ILs (BMImBF_4_ or BMImPF_6_, 10 mL, respectively) under vigorous stirring at room temperature. Then, the temperature was raised to 80 °C, and the mixture was stirred at this temperature for 1 h to obtain a uniform solution.

Next, compound **2** and **4** (2.0 mmol, nucleophile, respectively), anhydrous Na_2_CO_3_ (4.0 mmol), and the THE solvent system were added together, and then the mixture was further stirred at 80 °C for 8 h. The resulting mixture was carefully concentrated under reduced pressure to remove volatile solvents, and residue (IL-containing phase) was extracted by diethyl ether (3 × 100 mL). The left IL phase was then mixed with consumables (substrates, solvent, and base) for recycling. Organic layers were collected; washed with saturated NaHCO_3_ (100 mL), distilled H_2_O (100 mL), and brine (100 mL); dried over anhydrous Na_2_SO_4_; filtrated; and concentrated under reduced pressure. The crude product was purified by column chromatography (SiO_2_, 300 mesh; petroleum ether/ethyl acetate, 4/1, *v/v*; adding ten drops of triethylamine in eluent of 200 mL). The cross-coupling products (compound **3** or **5**) were obtained as a yellow oil and a white solid, respectively.

### 3.6. Characterization of Tert-butyl-3-(3-(methoxycarbonyl)phenyl)-1H-indazole-1-carboxylate (Compound **3**)

Yellow oil. ^1^H NMR (400 MHz, CDCl_3_) δ_H_, ppm: 1.74 (9H, s, *tert*-butyl), 3.95 (3H, s, methyl), 7.38 (1H, t, *J* = 8.0 Hz, Ar*H*), 7.54–7.61 (2H, m, Ar*H*), 7.97 (1H, d, *J* = 8.0 Hz, Ar*H*), 8.12 (1H, d, *J* = 8.0 Hz, Ar*H*), 8.19–8.22 (2H, m, Ar*H*), 8.66 (1H, t, *J* = 4.0 Hz, Ar*H*) (Appendix A). ^13^C NMR (101 MHz, CDCl_3_) δ_C_, ppm: 166.83, 149.40, 148.87, 141.12, 132.74, 132.50, 130.87, 130.39, 129.39, 129.09, 129.02, 124.19, 124.10, 121.33, 115.07, 85.18, 52.38, 28.28. Analysis calcd for C_20_H_20_N_2_O_4_: C, 68.18; H, 5.68; N, 7.95. Found: C, 68.69; H, 5.19; N, 7.39.

### 3.7. Characterization of Tert-butyl-3-(4-(methoxycarbonyl)phenyl)-1H-indazole-1- carboxylate (Compound **5**)

White solid. M.p., 74.0–75.0 °C. ^1^H NMR (400 MHz, CDCl_3_) δ_H_, ppm: 1.75 (9H, s, *tert*-butyl), 3.95 (3H, s, C*H*_3_), 7.38 (1H, t, *J* = 8.0 Hz, Ar*H*), 7.56 (1H, t, *J* = 8.0 Hz, Ar*H*), 7.96 (1H, d, *J* = 8.0 Hz, Ar*H*), 8.08–8.16 (2H, m, Ar*H*), 8.17–8.22 (3H, m, Ar*H*) (Appendix A). ^13^C NMR (101 MHz, CDCl_3_) δ_C_, ppm: 166.84, 149.33, 148.69, 141.16, 136.51, 130.68, 130.14, 129.05, 128.25, 124.25, 124.10, 121.27, 115.11, 85.27, 52.36, 28.26. Analysis calcd for C_20_H_20_N_2_O_4_: C, 68.18; H, 5.68; N, 7.95. Found: C, 68.79; H, 5.91; N, 7.34.

## 4. Conclusions

In conclusion, a series of silica gel-supported PdO_2_ nanoparticles were prepared as catalysts for Suzuki–Miyaura cross-coupling reactions leading to the C-3 modification of 1*H*-indazole. Ball milling appeared to be a powerful method to construct PdO_2_ nanoparticles with a size of 25–66 nm over silica gel by using various Pd(II) precursors as raw material. The use of Pd(OAc)_2_ and PdCl_2_(dppf) as synthetic materials maintained the morphology and internal structure of silica gel backbone basically, but the immobilization of PdCl_2_(dtbpf) changed the silica gel backbone to a large extent.

All the synthesized catalysts showed much higher outputs than their palladium precursors, emphasizing that the PdO_2_ nanoparticles immobilized over silica gel had high activity. Although the homogeneous palladium precursors showed an order of activity of PdCl_2_(dppf) > PdCl_2_(dtbpf) > Pd(OAc)_2_, the order of synthesized catalysts appeared to be C1 (from Pd(OAc)_2_) > C3 (from PdCl_2_(dtbpf)) > C2 (from PdCl_2_(dppf)), which conformed to the orders of the BET surface area and the acid amount of the synthesized catalysts, also indicating that the ligand of the palladium precursor showed important influence on the building activity of PdO_2_ nanoparticles.

In view of the much higher costs of PdCl_2_(dppf) and PdCl_2_(dtbpf) compared to Pd(OAc)_2_, this work obviously provided an efficient and inexpensive profile for the C-3 functionalization of 1*H*-indazole through Suzuki–Miyaura cross-coupling. Additionally, the recycling of synthesized catalysts in a toluene/water/ethanol mixed solvent showed satisfactory results, and C1 maintained its high activity during continuous use. On the other hand, the introduction of imidazolium ionic liquids as recycling media was ineffective, which may block the dispersion of substrates into catalytic centers (PdO_2_ nanoparticles). Evidently, this work will contribute to the exploration of 1*H*-indazole-based pharmaceuticals.

## Data Availability

We all authors would like to share our research data.

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
