# Peer review of "Mechanochemical Synthesis of PdO2 Nanoparticles Immobilized over Silica Gel for Catalytic Suzuki–Miyaura Cross-Coupling Reactions Leading to the C-3 Modification of 1H-Indazole with Phenylboronic Acids"

_molecules, 2023, doi:10.3390/molecules28207190_

Round 1

Reviewer 1 Report

[The reviewer is mainly concerned with the section of synthetic methods].

This manuscript of Yan Sun’s group presents the work on the titled reaction using one known iodinated substrate with two commercially available boronic acid nucleophiles to provide the known coupling products.  The reviewer considers that the major content of the work is focused on a mechanochemical study as described in the title. 

Besides the mechanochemical standpoints, explanations for well-investigated Suzuki-Miyaura cross-coupling parts should be more and more squeezed and/or omitted helped with suitable citations of reported reviews and the original papers.  In addition, a number of repetitive descriptions appear throughout the manuscript; more and more concise sentences are required.  Especially, Introduction and Conclusion parts is too long to smoothly comprehend the contents; the reader undertaking laborious efforts.  For example, the reviewer recommends that the backgrounds and this work (lines 120-146) should be visualized as a “Scheme” style.

Unfortunately, for the recycling study, the effect of greener ionic liquids is clearly negative (Figures 10 and 11). 

  On the whole, the reviewer recommends the publication in Molecules after major revision (introduction and synthesis sections) described.

<Comments and suggestions>

1.     Abstract: line 19; inert → less reactive

2.     Abstract: line 28; the abbreviation of “BET” should be specifically addressed.     

3.     Abstract: line 29; “In general” should be deleted, because this phrase is not suitable for the abstract. 

4.     Introduction: line 40; rearrangement → resonance structures

5.     Introduction: line 50; “Although being less stable and abundant” “Although being not only less stable and but also less abundant”? Please check it, because “less stable” and “abundant” is contradicted each other.   

6.     Introduction: line 63; scientists → synthetic organic chemists

7.     Introduction: lines 69-78; Too long. Three “1,10-phenanthroline”s are repetitive. 

8.     Introduction: line 108; “In practice” should be deleted, because it is meaningless.

9.     Introduction: lines 179-187; Due to the repetition, more concise description is needed. 

10.  2.5.1: Synthesis of Catalytic Substrate Preparation of Iodinated 1H-indazole Substrate

11.  lines 387; 1H-indazole 1H-indazole 1 or its IUPAC name (1)

12.  lines 388; 1H-indazole → the substrate 1

13.  lines 388-393; Three “cross coupling”s appeared. More concise expressions is needed, because this issue is the reported method (Ref. 43)

14.  2.5.1: Effect of Substrate → Reactions using Two Substrates   

15.  Tables 3 and 4; “Solvent” “Co-solvent”  

16.  Tables 3 and 4; Many descriptions in the footnote are repetitive, because these are addressed in the equation. 

17.  lines 406-409; “As known ……” The reviewer cannot understand the meaning. Is this consistent with  the reported results (Ref. 44)?       

18.  2.5.3: Effect of Catalyst → Screening of Catalysis        

19.  line 413; “proceeded well and” should be delated.

20.  lines 425-484; More and more concise descriptions are needed, because these contents are simply following the reported review (Ref. 13).  Two “Furthermore”s are repetitive.

21.  line 572; “were” → “are” due to a common style of the reaction mechanism.  

22.  In the experimental section, “Firstly, secondly, and thirdly”, “Furthermore”, “Moreover”, and Additionally” are not appropriate and deleted due to the common style.  Additionally, the sentence tense should be “passive”.

23.  lines 584-596; More and more concise descriptions are needed, because the reported content (Ref. 45).

24.  ines 671-698; “At first”, “In practice”, “Next” should be deleted.

As described in the reviewer's comments, extensive editing of English language required.

Author Response

For comments from Reviewer 1

Reviewer 1:

This manuscript of Yan Sun’s group presents the work on the titled reaction using one known iodinated substrate with two commercially available boronic acid nucleophiles to provide the known coupling products.  The reviewer considers that the major content of the work is focused on a mechanochemical study as described in the title.

√  First of all, we are really grateful for Reviewer 1’s time, evaluation and work on our submission. Thanks.

Yes, we agree that the major content of our work is focused on the ball milling synthesis of silica gel-supported PdO2 Nanoparticles. But in addition, we modestly suggest that the application of synthesized products in Suzuki–Miyaura Cross-Coupling leading to functionalization of 1H-Indazole deserves attentions, because a new, very available and inexpensive catalytic protocol is provided in this work. Hitherto, this transformation has been carried out by using very expensive Pd catalysts such as PdCl2(dppf) and PdCl2(dtbpf).

Besides the mechanochemical standpoints, explanations for well-investigated Suzuki-Miyaura cross-coupling parts should be more and more squeezed and/or omitted helped with suitable citations of reported reviews and the original papers.  In addition, a number of repetitive descriptions appear throughout the manuscript; more and more concise sentences are required.  Especially, Introduction and Conclusion parts is too long to smoothly comprehend the contents; the reader undertaking laborious efforts.  For example, the reviewer recommends that the backgrounds and this work (lines 120-146) should be visualized as a “Scheme” style.

√  Yes, we understand and agree with Reviewer 1’s comments.

At first, we accept that explanations for well-investigated Suzuki-Miyaura cross-coupling parts should be more and more squeezed and/or omitted helped with suitable citations of reported reviews and the original papers.

Therefore,

  • In Introduction, the original fifth, sixth and seventh paragraphs are merged into one paragraph, and revised as follows:

“Additionally, both Heck [12] and Suzuki-Miyaura cross-coupling [1] had been introduced to C-3 functionalization of indazole, and highly valuable results were obtained. In the light of the progresses made so far, it seemed that 1H-indazole can be employed as either nucleophile or electrophile in metal-catalyzed coupling reactions, proposing a much wider and more precise substrate scope for future pharmaceutical design. However, the catalysts employed so far were expensive and non-recyclable, and rection conditions were too tough to be amplified.

Among metal-catalyzed coupling reactions developed so far, Suzuki-Miyaura cross-coupling seemed to be a highly promising protocol for C-3 functionalization of 1H-indazole. At first, C-3 position of 1H-indazole showed very poor reactivity, causing many metal-catalyzed cross-couplings like Heck reaction less effective (yielding more by-products) [12]. However, C‒H bond of this position can be iodinated stoichiometrically without affecting the amino group of 1H-indazole, affording an electrophile for being coupled with organoboronic acids (nucleophile), which may realize C-3 functionalization of 1H-indazole [1,6].

Next, most Suzuki-Miyaura cross-coupling reactions could be accomplished below 80 oC within 3 h, without sensitivity to air or moisture, obviously decreasing operational risk and energy consumption [13]. This transformation usually did not release any toxic gases, which was advantageous to environmental protection [14]. Therefore, there is a great opportunity for the application of Suzuki-Miyaura cross-coupling reaction in C-3 functionalization of 1H-indazole.”

was changed to

“And meanwhile, both Heck [12] and Suzuki-Miyaura cross-couplings [1] had ever been introduced to C-3 functionalization of indazole, and significant results were obtained. Heck reaction often showed poorer efficiency for C-3 functionalization, because the coupling partner demanded a considerably irreversible protection of N-H group of 1H-indazole, otherwise leading to unwanted by-products [12]. In comparison, the suitable coupling partners of Suzuki-Miyaura cross-coupling such as organoboronic acids only needed a highly removable protection of N‒H group, like that from di-tert-butyldicarbonate ((Boc)2O) [1]. Furthermore, most Suzuki-Miyaura cross-couplings could be accomplished at mild temperature (lower than 80 oC) within a short time (3 h) [13], producing no toxic pollutants [14], which were comparable to Heck reactions [12].”.

  • In Introduction, the original eighth and ninth paragraphs are merged into one paragraph, and revised as follows:

“However, there were still several drawbacks on Suzuki-Miyaura cross-coupling reaction deserved attentions. First of all, it was widely reported that palladium catalysts usually showed much higher reactivity than nickel or copper ones, but palladium precursors (salts) were considerably expensive in synthesis of catalysts [13,14]. Secondly, synthesis of palladium catalysts often needed costly organic ligands, and the processes seemed sophisticated too [14]. At last, palladium catalysts were usually lack of effective profile for recovery, leading to high production cost, inconvenient purification of product, as well as environmental pollution [13,14].

In practice, therefore, a lot of attempts on immobilization of palladium into a variety of supports came into sight in order for recycling palladium catalysts during the past two decades. The support materials included polymer [15], polysiloxane [16], covalent organic frameworks (COFs) [17], metal-organic frameworks (MOFs) [18], as well as carbon materials [19]. Overall, the structure of one palladium complex immobilized over support had differed from its homogeneous counterpart, usually showing lower activity [14]. In rare cases, however, the support materials can be induced as active ligand so as to reshape the steric volume, size and porosity of palladium precursors, showing positive influence on catalytic yields of Suzuki-Miyaura cross-coupling, like the phosphine-polymeric palladium complex fixed on surface of silica [20], or polysiloxane-supported palladium [16]. Until now, the research on developing support materials along with effective immobilization method attracted wide and continuous attentions.”

was changed to

    “In practice, however, there were still some drawbacks on conducting Suzuki-Miyaura cross-coupling reaction, including high cost of palladium precursors (salts) [13], lengthy and costly synthesis of palladium catalysts [14], as well as the lack of profile for catalyst recovery [13,14]. Therefore, a lot of attempts on immobilization of palladium into various supports were carried out in order for simplifying catalyst synthesis and improving catalyst recovery. The support materials covered polymer [15], polysiloxane [16], covalent organic frameworks (COFs) [17], metal-organic frameworks (MOFs) [18], as well as carbon materials [19]. Overall, the structure of palladium complex immobilized over support usually differed from its homogeneous counterpart, giving lower activity [14]. But in rare cases, the support may play as active ligand reshaping the steric volume, size and porosity of palladium precursors, showing positive influence on catalytic outputs [16,20].”.

  • In Introduction, the original tenth paragraph is revised as follows:

“From another point of view, some recently developed approaches just like mechanochemistry emerged as a breakthrough of classical Suzuki-Miyaura cross-coupling, mainly focusing on the clean and energy-saving catalytic process [21], as well as preparation of highly active catalyst, along with effective recycling [22]. The mechanochemistry can be seen as chemical transformation propelled by mechanical energies involving friction, compression, forging, extrusion, and milling as well [23]. These means often provided enough energy for breaking, rebuilding or connecting chemical bonds, actually showing opportunities for establishment of new reactions or advanced materials [23].”

was changed to

“In addition to traditional catalytic and synthetic protocols, some currently developed approaches just like mechanochemistry looked like a real breakthrough for Suzuki-Miyaura cross-coupling, mainly regarding a clean and energy-saving catalytic reaction [21], as well as mechanochemical preparation of catalyst, along with effective recycling [22]. The mechanochemistry meant the chemical transformation propelled by mechanical energies involving friction, compression, forging, extrusion, and milling as well [23], which were high enough to break, rebuild or connect chemical bonds, showing opportunities for establishing new reactions or advanced materials [23].”.

  • In Introduction, the original eleventh paragraph is revised as follows:

“Nowadays, mechanochemistry had also become a highly attractive technology for a variety of chemical conversions, which may boost brand-new, highly chemo- or stereo-selective, and much safer reactions, because mechanical energies successfully circumvented drawbacks derived from classical powers [24]. For instance, classical heating such as artificially raising and decreasing temperature would produce unwanted by-products, and heating a closed reactor increased risk of explosion [23]. Secondly, it was reported that 85% of chemicals applied in pharmaceutical industry were solvents, but even if recovered, 20‒50% of solvents were lost by evaporation [24]. However, mechanochemical reactions needed marginal solvents or none at all, which saved a huge amount of solvent, leading to much greener and cheaper processes [24].”

was changed to

“On one hand, mechanochemistry appeared to be a highly attractive technology for achieving a variety of chemical conversions, which may boost brand-new, highly chemo- or stereo-selective, and much safer reactions, because mechanical energies successfully circumvented drawbacks derived from classical powers [24]. For instance, classical heating such as artificially raising and decreasing temperature would produce unwanted by-products, and heating a closed reactor increased risk of explosion [23]. Secondly, it was reported that 85% of chemicals applied in pharmaceutical industry were solvents, but even if recovered, 20‒50% of solvents were lost by evaporation [24]. However, mechanochemical reactions needed marginal solvents or none at all, which saved a huge amount of solvent, leading to much greener and cheaper processes [24].”.

  • In Introduction, the original twelfth paragraph is revised as follows:

“Furthermore, mechanochemistry emerged as a highly efficient methodology for synthesis of micro- or nano-sized catalysts, nanomaterials, or other submicroscopic materials, as compared to conventional synthetic or immobilizing profiles, which not only demanded multi-step processes, heating or introduction of hazardous and expensive reagents, but also gave less active catalysts (materials) [23].”

was changed to

    “On the other hand, mechanochemistry contributed to the synthesis or formation of micro- or nano-sized catalysts, nanomaterials, or other submicroscopic materials with great porosity [23]. In comparison, conventional synthetic profiles not only demanded lengthy processes, heating, or hazardous and expensive reagents, but also gave amorphous catalysts (materials) with less activity [23].”.

Second, we have carefully checked the whole manuscript, and revised the repetitive descriptions as follows:

  • In Introduction, the original fourteenth paragraph is revised as follows:

    “On the basis of these findings, it seemed that mechanochemistry (ball milling) may disperse catalytic components into nanosized particles over porous support materials without depressing porosity of supports.” was deleted.

Next, the conclusion part is revised as follows:

  • In the first paragraph of Conclusions,

“On the basis of full characterizations, it can be seen that the use of Pd(OAc)2 and PdCl2(dppf) as synthetic material would maintain the morphology and internal structure of silica gel backbone basically, but immobilization of PdCl2(dtbpf) changed those of silica gel to a large extent.”

was changed to

“The use of Pd(OAc)2 and PdCl2(dppf) as synthetic material would maintain the morphology and internal structure of silica gel backbone basically, but immobilization of PdCl2(dtbpf) changed those of silica gel to a large extent.”.

  • The second and third paragraphs of Conclusions were revised and divided into two new paragraphs as follows:

“All synthesized palladium catalysts showed much higher outputs for Suzuki–Miyaura cross-coupling than their palladium precursors, emphasizing the PdO2 nanoparticles immobilized over silica gel had high catalytic activity. Although homogeneous palladium precursors showed an order of catalytic activity as PdCl2(dppf) > PdCl2(dtbpf) > Pd(OAc)2, the order of synthesized palladium catalysts appeared to be C1 (from Pd(OAc)2) > C3 (from PdCl2(dtbpf)) > C2 (from PdCl2(dppf)), indicating ligand of palladium precursor showed important influence on building activity of PdO2 nanoparticles. In view of the much higher costs of PdCl2(dppf) and PdCl2(dtbpf) compared to Pd(OAc)2, this work obviously provided an efficient and inexpensive profile for C-3 functionalization of 1H-indazole through Suzuki–Miyaura cross-coupling.

Taking into account results of characterization, the catalytic activity order of C1 > C3 > C2 can be attributed to both BET surface area and acid amount of synthesized catalyst. The higher surface area and acidity realized, the better catalytic outputs were obtained. Furthermore, recycling of synthesized catalysts in a toluene/water/ethanol mixed solvent showed satisfactory results, and C1 maintained its high activity during continuous uses. However, introduction of imidazolium ILs as recycling media seemed ineffective, which blocked the dispersion of substrates into catalytic centers (PdO2 nanoparticles).”

was changed to

“All synthesized palladium catalysts showed much higher outputs than their palladium precursors in Suzuki–Miyaura cross-coupling, emphasizing the PdO2 nanoparticles immobilized over silica gel had high catalytic activity. Although homogeneous palladium precursors showed an order of catalytic activity as PdCl2(dppf) > PdCl2(dtbpf) > Pd(OAc)2, the order of synthesized palladium catalysts appeared to be C1 (from Pd(OAc)2) > C3 (from PdCl2(dtbpf)) > C2 (from PdCl2(dppf)), which conformed to the orders found in BET surface area and acid amount of synthesized catalysts, also indicating ligand of palladium precursor showed important influence on building activity of PdO2 nanoparticles.

In view of the much higher costs of PdCl2(dppf) and PdCl2(dtbpf) compared to Pd(OAc)2, this work obviously provided an efficient and inexpensive profile for C-3 functionalization of 1H-indazole through Suzuki–Miyaura cross-coupling. Additionally, recycling of synthesized catalysts in a toluene/water/ethanol mixed solvent showed satisfactory results, and C1 maintained its high activity during continuous uses. But on the other hand, the introduction of imidazolium ionic liquids as recycling media was ineffective, which may block the dispersion of substrates into catalytic centers (PdO2 nanoparticles).”.

  • The last paragraph of Conclusions was deleted.

Unfortunately, for the recycling study, the effect of greener ionic liquids is clearly negative (Figures 10 and 11). 

√  Yes, according to the catalytic results obtained in this work, the introduction of two imidazolium ionic liquids as greener solvent is negative.

On the whole, the reviewer recommends the publication in Molecules after major revision (introduction and synthesis sections) described.

√  Yes, we agree with this comment. Additionally, we will carefully polish and refine the whole manuscript at the end of this revision.

   Thanks for very helpful instructions and comments.

<Comments and suggestions>

  1. Abstract: line 19; inert → less reactive

√  Yes, in Abstract, line 19,

    “inert” was changed to “less reactive”.

  1. Abstract: line 28; the abbreviation of “BET” should be specifically addressed.

√  Yes, in Abstract, line 28,

“(Brunauer–Emmett–Teller)” was added.

  1. Abstract: line 29; “In general” should be deleted, because this phrase is not suitable for the abstract. 

√  Yes, in Abstract, line 28,

   “In general,” was deleted.

  1. Introduction: line 40; rearrangement → resonance structures

√  Yes, in Introduction, line 40,

   “due to rearrangement of C=N or N=N bonds on pyrazole nuclei of indazole” was changed to “due to the resonance structures of pyrazole nuclei of indazole”.

  1. Introduction: line 50; “Although being less stable and abundant” → “Although being not only less stable and but also less abundant”? Please check it, because “less stable” and “abundant” is contradicted each other. 

√  Yes, we accept this comment, in Introduction, line 40,

“Although being less stable and abundant than 1H-tautomer,” was changed to “Although being less stable and fewer than 1H-tautomer,”.

  1. Introduction: line 63; scientists → synthetic organic chemists

√  Yes, we accept this comment, in Introduction, line 63,

   “scientists” was changed to “synthetic organic chemists”.

  1. Introduction: lines 69-78; Too long. Three “1,10-phenanthroline”s are repetitive.

√  Yes, we accept this comment, in Introduction, line 69‒78,

    “, where 1,10-phenanthroline played a key role in activation of palladium catalyst” was deleted.

  1. Introduction: line 108; “In practice” should be deleted, because it is meaningless.

√  Yes, we accept this comment, in Introduction, line 108,

   “In practice, ” had been deleted according to previous revisons.

  1. Introduction: lines 179-187; Due to the repetition, more concise description is needed.

√  Yes, we accept this comment, in Introduction, lines 179‒187,

   “It was anticipated that ILs not only showed great activity of immobilization, but also exhibited eco-friendly properties just like non-volatility [29].” was deleted.

   Furthermore, “At first, palladium(II) acetate and two ferrocene-based palladium(II) complexes were respectively immobilized into silica gel (300 mesh, also used in column chromatography) by ball milling. After characterizations, the catalysts were employed in Suzuki-Miyaura cross-coupling of amino-protected 3-iodo-1H-indazole with phenylboronic acids. At last, the catalyst recycling was also tested with or without ILs. This work may show values on the design of new palladium nanocatalysts as well as on the manufacture of versatile 1H-indazole-based pharmaceuticals.” was changed to

    “In practice, various palladium(II) precursors were immobilized into silica gel through ball milling. The catalysts obtained were then employed in Suzuki-Miyaura cross-coupling of amino-protected 3-iodo-1H-indazole with phenylboronic acid. Catalyst recycling was also tested with or without ILs. This work may show values on the manufacture of 1H-indazole-based pharmaceuticals.”.

  1. 2.5.1: Synthesis of Catalytic Substrate → Preparation of Iodinated 1H-indazole Substrate

√  Yes, we accept this comment, in the title of Sect. 2.5.1.,

   “Synthesis of Catalytic Substrate” was changed to “Preparation of Iodinated 1H-indazole Substrate”.

  1. lines 387; 1H-indazole → 1H-indazole 1 or its IUPAC name (1)

√  Yes, we accept this comment, therefore,

   In Sect. 2.5.1., “1H-indazole” was changed to “1H-indazole (compound 1)”.

  1. lines 388; 1H-indazole → the substrate 1

√  Yes, we accept this comment, therefore,

    In Sect. 2.5.1., “1H-indazole” was changed to “the substrate 1”.

  1. lines 388-393; Three “cross coupling”s appeared. More concise expressions is needed, because this issue is the reported method (Ref. 43)

√  Yes, we accept this comment, therefore,

   In Sect. 2.5.1.,

   “Suzuki-Miyaura cross-coupling” was changed to “Suzuki-Miyaura reaction”.

   “After the cross-coupling reaction,” was changed to “After reaction,”.

  1. 2.5.1: Effect of Substrate → Reactions using Two Substrates

√  Yes, we accept this comment, therefore,

   In the title of Sect. 2.5.2.,

   “Effect of Substrate” was changed to “Reactions Using Two Substrates”.

  1. Tables 3 and 4; “Solvent” → “Co-solvent”

√  Yes, we accept this comment, therefore,

   In Tables 3‒4,

   “Solvent” on the first line of each Table was changed to “Solvent(/Co-solvent)”;

   And meanwhile, in annotation a of each Table,

   “Co-solvent” was added.

  1. Tables 3 and 4; Many descriptions in the footnote are repetitive, because these are addressed in the equation.

√  Yes, we accept this comment, therefore,

   In annotation a of each Table,

   “a Reaction conditions: compound 1 (2.0 mmol), compound 2 (2.0 mmol), Pd catalyst (C0, equal Si as C1; C1‒C3, 2‒4 mol% Pd, based on 1), THE solvent system (Toluene, 8 mL; H2O, 4 mL; EtOH, 2 mL), Na2CO3 (2.0‒4.0 mmol), ILs (BMImPF6 or BMImBF4, 10 mL; co-solvent), 40‒80 oC, 8 h.”

was changed to

a Reaction conditions as in equation of this Table and Sect. 3.5., Pd catalyst (C0, equal Si as C1, based on XPS results of Table 1; C1‒C3, 2‒4 mol% Pd, based on 1).”.

  1. lines 406-409; “As known ……” The reviewer cannot understand the meaning. Is this consistent with  the reported results (Ref. 44)? 

√  Yes, we accept this comment.

   After carefully reading Ref. 44 (Wang, J.; Zheng, W.; Ding, L.; Wang, Y. Computational study on C–B homolytic bond dissociation enthalpies of organoboron compounds. New J. Chem. 2017, 41, 1346–1362), it can be clearly seen that methoxycarbonyl is an electron withdrawing group (EWG), as shown in Pages 1351‒1352:

Furthermore, when methoxycarbonyl is ortho-substituted, meta-substituted or para-substituted on phenyl of organoboronic acid, the homolytic bond dissociation enthalpy (BDE) of the C–B bond on three positions showed an order as ortho-substituted < para-substituted < meta-substituted, meaning the cleavage of C-B bond of compound 4 needed less energy than that of compound 3. The direct reference is Ref. 44, Page 1354:

Therefore, in the first paragraph of Sect. 2.5.2,

“As known, methoxycarbonyl group appeared to be an o- and p-orienting substituent on phenyl, and meanwhile, the nucleophilicity of phenyl boronic acid (compounds 2 and 4) was derived from the negative charge at the carbon of C-B bond [44]. Therefore, the anion stemming from the para isomer (compound 4) seemed more stable than that from meta isomer (compound 2) in catalytic Suzuki–Miyaura cross-coupling, leading to better conversions (Tables 3‒4).”

was changed to

    “First of all, it was previously confirmed that methoxycarbonyl is an electron withdrawing group (EWG) [44]. Secondly, when methoxycarbonyl was ortho-, meta- or para-oriented on phenylboronic acid, the homolytic bond dissociation enthalpy (BDE) of the C–B bond showed an order as ortho- < para- < meta-oriented, which was also the order of C–B bond strength [44], meaning the cleavage of C-B bond of compound 4 needed less energy than that of compound 3.”

  1. 2.5.3: Effect of Catalyst → Screening of Catalysis

√  Yes, we accept this comment.,

   Therefore, in the title of Sect. 2.5.3.,

   “Effect of Catalyst” was changed to “Screening of Catalysts”.

  1. line 413; “proceeded well and” should be delated.

√  Yes, we accept this comment.,

   Therefore, in the first paragraph of Sect. 2.5.3.,

   “Overall, most Suzuki–Miyaura cross-coupling reactions proceeded well and were accomplished within 8 h having no detectable by-products.” was changed to “Overall, most Suzuki–Miyaura cross-coupling reactions were accomplished within 8 h having no detectable by-products.”.

  1. lines 425-484; More and more concise descriptions are needed, because these contents are simply following the reported review (Ref. 13).  Two “Furthermore”s are repetitive.

√  Yes, we accept this comment.,

Therefore, in the fifth paragraph of Sect. 2.5.3.,

“According to previous reports, there were three steps in typical Suzuki–Miyaura cross-coupling mechanism, including oxidative addition, transmetallation, and reductive elimination as well [13]. At first, various Pd(II) precursors were activated in situ into catalytically active Pd(0) species, which was then oxidatively added by phenyl halide (PhX, like compound 1), yielding the Pd intermediate [PhPdXLn]. Subsequently, the halogen anion (X-) of [PhPdXLn] was substituted with the pre-added base (OH-), leading to [PhPdOHLn] intermediate, which would be quickly reacted with organoboronic acid (Ph’B(OH)2, like compounds 2 or 4), providing the diphenyl complex [PhPdPh’Ln], which was summarized as transmetallation. Lastly, reductive elimination of the diphenyl complex gave the biphenyl product Ph–Ph’ (compounds 3 or 5), along with Pd(0) for recycling. Herein, it seemed that ligand (L) of Pd(II) precursor showed important influence on stabilizing the Pd intermediate [PhPdXLn] in the step of oxidative addition, and obviously acetate anion was much less effective than both dppf and dtbpf in this work.”

was changed to

“There were three steps in Suzuki–Miyaura cross-coupling, including oxidative addition, transmetallation, and reductive elimination [13]. The ligand of palladium precursor played a key role in oxidative addition, directly affecting the catalytic outputs [13].”.

And furthermore, this paragraph was merged to the last paragraph.

  1. line 572; “were” → “are” due to a common style of the reaction mechanism.

√  Yes, we accept this comment.,

   Therefore, in Sect. 2.6.,

   The past indefinite was changed to the present tense.

  1. In the experimental section, “Firstly, secondly, and thirdly”, “Furthermore”, “Moreover”, and Additionally” are not appropriate and deleted due to the common style.  Additionally, the sentence tense should be “passive”.

√  Yes, we accept this comment.,

Therefore,

  • In the first paragraph of Sect. 3.1.,

“(PdCl2(dppf), 98%, Pd > 13%)” was changed to “(PdCl2(dppf), purity of 98%, content of Pd > 13%)”;

  • In the second paragraph of Sect. 3.1.,

“Two imidazolium ILs, BMImX (BMIm+ = 1-n-butyl-3-methylimidazolium; X- = BF4-, PF6-; both 98%) were commercially available from Alfa Aesar (part of Thermo Fisher Scientific, Thermo Fisher Scientific (China) Co., Ltd., Shanghai, China).”

was changed to

“Two imidazolium ILs, abbreviated as BMImX (BMIm+ = 1-n-butyl-3-methylimidazolium; X- = BF4-, PF6-; both purities of 98%) were bought from Alfa Aesar (Thermo Fisher Scientific (China) Co., Ltd., Shanghai, China).”

  • In the first paragraph of Sect. 3.2.,

“At first,” was changed to “The”.

  • In the first paragraph of Sect. 3.2.,

“Secondly, X-ray photoelectron spectroscopy (XPS) was performed on a Kratos Axis Ultra DLD (Kratos Co., Ltd., Manchester, UK), employing monochromatic Al K α X-rays (1486.6 eV) as lighting source. The binding energy scale was calibrated by using C 1s peak at 284.8 eV as standard. The peaks were fitted by using Gaussian–Lorentz (G/L) product function with 30% Lorentzian ratio.”

was changed to

“The X-ray photoelectron spectroscopy (XPS) was performed on a Kratos Axis Ultra DLD (Kratos Co., Ltd., Manchester, UK), whose lighting source selected monochromatic Al K α X-rays (1486.6 eV). During data processing, binding energy scale was first calibrated by setting C 1s peak at 284.8 eV. The peaks were then fitted by using Gaussian–Lorentz (G/L) product function with 30% Lorentzian component.”.

  • In the second paragraph of Sect. 3.2.,

“Thirdly, the wide-angle (2θ = 10–80o) X-ray diffractions were employed to test the crystallinity of synthesized catalyst on a Philips X’Pert Pro diffractometer (PANalytical B.V. Co., Ltd., Almelo, Holland), using Cu-Kα radiation (λ = 1.5418 Å) with interval of 0.05o s-1.”

was changed to

“The wide-angle (2θ = 10–80o) X-ray diffractions were performed on a Philips X’Pert Pro diffractometer (PANalytical B.V. Co., Ltd., Almelo, Holland), whose X-ray source was Cu-Kα radiation (λ = 1.5418 Å), along with diffraction angle interval of 0.05o s-1.”.

  • In the second paragraph of Sect. 3.2.,

“Next, the porosity parameters including BET surface area, total pore volume, micropore volume and pore size were measured on Micromeritics ASAP 2020 (Micromeritics Instruments Corporation, Norcross, Atlanta, GA, USA), where the N2 adsorption isotherms were obtained at 77.35 K. Synthesized catalysts were degassed in vacuum at 150 oC before testing. Herein, surface area was calculated by using the multi-point Brunauer–Emmett–Teller (BET) method to adsorption data with relative pressure P/P0 of 0.06–0.3. Total pore volume was obtained from N2 adsorbed at P/P0 = 0.97. Both pore volume and pore size were calculated by employing the Barrett–Joyner–Halenda (BJH) method.”

was changed to

“The porosity parameters including BET surface area, total pore volume, micropore volume and pore size were determined on Micromeritics ASAP 2020 (Micromeritics Instruments Corporation, Norcross, Atlanta, GA, USA). The nitrogen adsorption isotherms were measured at 77.35 K, and samples were degassed in vacuum at 150 oC before testing. The surface area was calculated by using the multi-point Brunauer–Emmett–Teller (BET) method in the light of adsorption data with relative pressure P/P0 at 0.06–0.3. Total pore volume was derived from nitrogen adsorbed at P/P0 = 0.97. Both pore volume and pore size were calculated by using the Barrett–Joyner–Halenda (BJH) method.”.

  • In the third paragraph of Sect. 3.2.,

“Furthermore, the acid amounts (both Lewis and Brönsted acidity) of synthesized catalysts can be determined according to literature with modifications [47]. Overall, a n-butylamine titration was employed in association with coeruleum bromocresolis as indicator. In practice, the sample (300 mg) was first combined with n-butylamine (25.00 mL, 0.05 mol L-1 in toluene solution) in a conical flask (250 mL). After shaking for 5 min under a cover, 2-proponal (100 mL) and coeruleum bromocresolis (one drop, diluted solution) were added together. The resulting solution was then titrated with HCl solution (0.025 mol L-1 in water), and the end point of titration was determined when solution color changed from blue to yellow. The acid amount of sample was evaluated as the amount of n-butylamine adsorbed, being calculated by subtraction of the residues (n-butylamine) in the solution (determined by HCl titration) from the pre-added total n-butylamine.”

    was changed to

“The acid amounts including both Lewis and Brönsted acidities of synthesized samples were determined by n-butylamine titration in association with coeruleum bromocresolis as indicator. This protocol was promoted previously [47]. The sample (300 mg) was first combined with n-butylamine solution (25.00 mL, 0.05 mol L-1 in toluene) into a conical flask (250 mL). After shaking for 5 min under cover, 2-proponal (100 mL) and coeruleum bromocresolis (one drop, diluted solution) were added consequently. The solution obtained was then titrated with HCl solution (0.025 mol L-1 in water), and the end point of titration was judged when solution color was changed from blue to yellow. The acid amount of sample was evaluated as the amount of n-butylamine adsorbed, through subtraction of the residues (n-butylamine) in the solution (determined by HCl titration) from the pre-added total n-butylamine.”.

  • In the fourth paragraph of Sect. 3.2.,

“Moreover, FT-IR spectra of synthesized catalysts were obtained when being dispersed in potassium bromide pellets on Bruker Tensor 27 spectrometer. UV–Vis spectroscopy was taken on UV 1800, Shimadzu. Thermogravimetric analysis (TGA) was performed on NETZSH TG 209C featuring TASC 414/4 controller under N2 protection, with a heating rate of 10 °C/min at 30–600 °C.”

was changed to

“FT-IR spectra were obtained on Bruker Tensor 27 spectrometer when samples were dispersed into potassium bromide pellets. UV–Vis spectroscopy was obtained on UV 1800, Shimadzu. Thermogravimetric analysis (TGA) was performed on NETZSH TG 209C, having TASC 414/4 controller under nitrogen protection, with a heating rate of 10 °C/min at 30–600 °C.”.

  • In the fifth paragraph of Sect. 3.2.,

“Additionally, scanning electron microscopy (SEM) was taken on JEOL JSM-6700F (JEOL Ltd., Tokyo, Japan) at 20.0 kV without Au coating. Transmission electron microscopy (TEM) was tested on JEOL JEM-200CX (JEOL Ltd., Tokyo, Japan) at 120 kV.”

was changed to

    “The scanning electron microscopy (SEM) was taken on JEOL JSM-6700F (JEOL Ltd., Tokyo, Japan) at 20.0 kV in the absence of Au coating. Transmission electron microscopy (TEM) was tested on JEOL JEM-200CX (JEOL Ltd., Tokyo, Japan) at 120 kV.”. And this paragraph was merged into the last paragraph.

  • In the last paragraph of Sect. 3.2.,

“As for the characterizations of catalytic products, the melting points of synthesized compounds were tested on WRR-Y melting point apparatus, Shanghai INESA Physico-Optical Instrument Co., Ltd. (Shanghai, China). 1H NMR was detected on Bruker ADVANCE III instrument (400 MHz), and 13C NMR still on Bruker ADVANCE III (101 MHz). The C, H, N elemental analyses were carried out on an Elementar VarioEL III instrument (Langenselbold, Germany).”

was changed to

    “As for the characterizations of catalytic products, the melting points were tested on WRR-Y melting point apparatus (Shanghai INESA Physico-Optical Instrument Co., Ltd., Shanghai, China). 1H NMR was detected on Bruker ADVANCE III instrument (using 400 MHz practically), and 13C NMR still on Bruker ADVANCE III (using 101 MHz practically). The C, H, N elemental analyses were carried out on an Elementar VarioEL III instrument (Langenselbold, Germany).”.

  1. lines 584-596; More and more concise descriptions are needed, because the reported content (Ref. 45).

√  Yes, we accept this comment.,

   Therefore,

  • in the second paragraph of Sect. 2.6.,

   “After oxidative addition of TS1 with compound 1, the resulting TS2 was further coordinated by substituted phenyl anion (TS2‒TS3, Figure12). Herein, the hydrolysis of Na2CO3 (base) provided OH-, whose nucleophilic attack to boron center of compound 4 gave substituted phenyl anion, building transmetallation. Furthermore, TS3 underwent reductive elimination, yielding not only target product (compound 5), but also TS1 for recycling (TS3‒TS1, Figure 12).”

   was changed to

   “The resulting TS2 was further coordinated by substituted phenyl anion (TS2‒TS3, Figure12). Herein, the hydrolysis of Na2CO3 (base) provided OH-, whose nucleophilic attack to boron of compound 4 gave phenyl anion. Furthermore, TS3 underwent reductive elimination, yielding not only target product (compound 5), but also TS1 for recycling (TS3‒TS1, Figure 12).”.

  • In the third and forth paragraphs of Sect. 2.6.,

“In comparison, there are several differences between the water-soluble Pd(II) complex-catalyzed (homogeneous) and PdO2 nanoparticle-catalyzed (heterogeneous) cross-couplings. Firstly, in homogeneous catalysis including IL-supported ones, ligand of Pd(II) precursor plays a key role in stabilizing intermediate during oxidative addition, but this effect becomes marginal on surface of PdO2 nanoparticles, because the surficial hydroxyl groups turns out to be great substitutes for those ligands, even contributing to the following transmetallation and reductive elimination.

Secondly, it was reported that the immobilization of homogeneous Pd(II) complexes into ILs may enhance more than depress both oxidative addition and transmetallation, due to full dispersion of Pd2+ along with its ligand into ILs [45]. However, the dispersion of PdO2 nanoparticles immobilized over silica gel into ILs is proved much less effective, because surface of PdO2 nanoparticles are saturated and internal pores are blocked by ILs, leading to poor approach of substrates to catalytically centers.”

was changed to

“In comparison, ligand of palladium(II) precursor stabilizes intermediate during oxidative addition, but this effect becomes marginal on surface of PdO2 nanoparticles, where surficial hydroxyl groups turns out to be great substitutes for those ligands. On the other hand, although immobilization of homogeneous palladium(II) complexes into ILs enhanced both oxidative addition and transmetallation [45], the dispersion of PdO2 nanoparticles immobilized over silica gel into ILs was not successful, probably because surface of PdO2 nanoparticles are saturated and internal pores are blocked by ILs.”.

  1. ines 671-698; “At first”, “In practice”, “Next” should be deleted.

√  Yes, we accept this comment.,

   Therefore, in Sect. 3.4.,

   “At first”, “In practice” and “Next” were deleted.

As described in the reviewer's comments, extensive editing of English language required.

√  Yes, we accept this comment.,

   Therefore, in addition to the above revisions, we will further polish the English language at the end of the whole revison.

That’s all for revisions according to Reviewer 1.

We all authors are really grateful for Reviewer 1’s very helpful comments and very kind reminding. Thanks!

Reviewer 2 Report

The manuscript reports the synthesis of silica gel-supported PdO2 nanoparticles by ball milling of silica gel with divalent palladium precursors; the obtained materials were used as catalyst for Suzuki–Miyaura cross-coupling of 1H-indazole derivative with phenylboronic acid.

The following are the observations to the manuscript:

- Line 24. Include information about the number of catalyst recycle and the variation of conversion and selectivity with catalyst reuse.

- Lines 27 and 28. The text “which was caused by BET surface area and acidity” is not clear about which materials are referred to.

- The abstract should include some details about the mechanochemical synthesis of the catalyst.

- Figure 1. What is the meaning of the numbers in the molecules and why not all the molecules have them?

- It is not specified what is C0, C1 or C3 materials.

- Figure 4. There are some XRD peaks that were not identified at around 36°, 43°, 53° and 68°. What is SiP?

- Although Scheme 1 is frequently cited, that scheme was not included in the manuscript.

- Why the maximum temperature used in TGA analysis was 600°C? What happens with the materials at higher temperatures? It seems that the loss of matter had not finished.

- Include a comparison of the found catalytic activity with other reported systems.

- Describe the procedure that was followed for activation of the catalyst before reusing it.

- Because the presence of water in THE solvent system, were two phases present in the system?

- What is the role in the reaction of each of the components of THE solvent system?

- Line 614. Gove some information about the used purification procedure of the solvents.

-As the milling beads were of ZrO2, was identified if some material of the beads were deposited on the final catalyst because of the milling and if was there any catalytic effect?

- Line 652. What do the authors mean with “under N2 protection”?

- Line 665. The word “respectively” is not necessary.

- Line 666. What was the used stirring rate?

- Line 725. The “THE solvent system” was described in line 705, it is not necessary to detailed again.

Author Response

For comments from Reviewer 2

Reviewer 2:

The manuscript reports the synthesis of silica gel-supported PdO2 nanoparticles by ball milling of silica gel with divalent palladium precursors; the obtained materials were used as catalyst for Suzuki–Miyaura cross-coupling of 1H-indazole derivative with phenylboronic acid.

√ First of all, we all authors are really grateful for Reviewer 2’s time, evaluation and work on our submission. Thanks.

The following are the observations to the manuscript:

- Line 24. Include information about the number of catalyst recycle and the variation of conversion and selectivity with catalyst reuse.

√  Yes, we accept this comment.,

Therefore, in Line 24,

“All synthesized catalysts showed much higher cross-coupling yields than their palladium precursors, along with great recyclability.” was changed to “All synthesized catalysts showed much higher cross-coupling yields than their palladium precursors, which could be also reused for three times without losing high activity and selectivity in a toluene/water/ethanol mixed solvent.”.

- Lines 27 and 28. The text “which was caused by BET surface area and acidity” is not clear about which materials are referred to.

√  Yes, we accept this comment.,

   Therefore, in Lines 27‒28,

   “which was caused by BET (Brunauer–Emmett–Teller) surface area and acidity.” was changed to ““which conformed to the orders of BET (Brunauer–Emmett–Teller) surface areas and acidities of these catalysts.”.

- The abstract should include some details about the mechanochemical synthesis of the catalyst.

√  Yes, we accept this comment.,

   Therefore, in Abstract,

   “were prepared by ball milling of silica gel with divalent palladium precursors” was changed to “were prepared by ball milling of silica gel with divalent palladium precursors in water”.

- Figure 1. What is the meaning of the numbers in the molecules and why not all the molecules have them?

√  Yes, we understand and respect this comment,

   In Figure 1, the 1H-indazole backbone was highlighted in red, and the number of atom was in blue. This arrangement would emphasize the importance of 1H-indazole-derived pharmaceuticals. Since that Figures 1a‒1c had shown the structures of three tautomers of indazole clearly, we consider that Figures 1d‒1f would not arouse misunderstandings. Thanks for reminding.

- It is not specified what is C0, C1 or C3 materials.

√  Yes, we understand and accept this comment, this is a very helpful reminding to us. Thanks. Actually, this submission included Scheme 1, which showed the synthesis of catalysts. However, Scheme 1 was missed during submission process due to my carelessness. Sorry for this mistake. The Scheme 1 was added in this revision.

Scheme 1. Synthesis of catalysts.

Therefore, in the first paragraph of Sect. 2.1.,

“The synthesis of catalysts was shown in Scheme 1.” was added.

    And meanwhile, Scheme 1 was added below the first paragraph of Sect. 2.1.

- Figure 4. There are some XRD peaks that were not identified at around 36°, 43°, 53° and 68°. What is SiP?

√  Yes, we understand and respect this comment, and we are really grateful for Reviewer’s very kind and helpful reminding.

In XRD data analysis, after matching of many times in association with software, it can be found that PdO2 (palladium oxide, PDF No. 34-1101) probably appeared on C1‒C3 (Figures 4b‒4d). However, other peaks occurred at 36°, 43°, 53° and 68° were not identified.

On the other hand, due to the same analysis, the phase of SiP may appear on XRD of C3 (Figure 4d).

The above deductions were supported by other characterizations such as XPS.

Therefore, according to Reviewer’s instruction and comments,

  • In the fourth paragraph of Sect. 2.1.,

“In order to further understand the composition of Pd species that formed on C1‒C3, the wide-angle (2θ = 10o–80o) X-ray diffraction (XRD) was employed as shown in Figure 4. In practice, there was a series of diffractions occurred at 2θ = 35.15o, 58.15o and 67.15o (grey cubes, Figure 4b), corresponding to 101, 220, 112 indices derived from PdO2 (palladium oxide, PDF No. 34-1101). The diffractions of PdO2 could also be found in wide-angle XRD of C2‒C3 (grey cubes, Figures 4c‒4d vs. 4b). Therefore, it seemed that PdO2 emerged as the metal-containing phase after ball milling, no matter which kind of divalent palladium precursor was used (Scheme 1).”

was changed to

“In order to further understand the composition of Pd species on C1‒C3, the wide-angle (2θ = 10o–80o) X-ray diffraction (XRD) was performed as shown in Figure 4. There was a series of diffractions occurred at 2θ = 35.15o, 58.15o and 67.15o (grey cubes, Figure 4b), probably corresponding to 101, 220, 112 indices derived from PdO2 (palladium oxide, PDF No. 34-1101). The diffractions of PdO2 may also be found in wide-angle XRD of C2‒C3 (grey cubes, Figures 4c‒4d vs. 4b). Therefore, it seemed that PdO2 probably emerged as the metal-containing phase after ball milling, no matter which kind of divalent palladium precursor was used (Scheme 1).”.

  • In the seventh paragraph of Sect. 2.1.,

“The C2 showed a similar XRD contour as compared to C1 (Figures 4c vs. 4b), but the diffraction of grunerite was detected (dark cube, Figure 4c; PDF No. 44-1401, Fe7Si8O22(OH)2), probably indicating dppf ligand was decomposed and transformed into Fe-Si mixed oxide during ball milling (Scheme 1). The C3 showed a higher binding energy than C0‒C2 (Figures S1d vs. S1a‒S1c), probably indicating some new Si-containing phases appeared during immobilization (Scheme 1). On one hand, in addition to PdO2, the diffraction of SiP appeared on XRD of C3 (hollow cube, Figure 4d; PDF No. 29-1133, SiP). On the other hand, there were two peaks occurred at 129.2 and 122.8 eV on P 2p region of C3 (Figure S2, Sect. S2, Supplementary Materials), probably corresponding to 2p1/2 and 2p3/2 photoelectrons of P4- [35], which were both lower than those of phosphorous with high valence coming from P‒O or P=O bonds [36].”

was changed to

“The C2 showed a similar XRD contour as C1 (Figures 4c vs. 4b), but the diffraction of grunerite might be detected (dark cube, Figure 4c; PDF No. 44-1401, Fe7Si8O22(OH)2), probably indicating dppf ligand was decomposed and transformed into Fe-Si mixed oxide during ball milling (Scheme 1). The C3 showed a higher binding energy of Si 2p photoelectron than C0‒C2 (Figures S1d vs. S1a‒S1c), indicating some new Si-containing phases appeared during immobilization (Scheme 1). On one hand, in addition to PdO2, the 002 diffraction of SiP may appear on XRD of C3 (hollow cube, Figure 4d; PDF No. 29-1133, SiP). On the other hand, there were two peaks occurred at 129.2 and 122.8 eV on P 2p region of C3 (Figure S2, Sect. S2, Supplementary Materials), probably corresponding to 2p1/2 and 2p3/2 photoelectrons of P4- [35], which were both lower than those of phosphorous with high valence coming from P‒O or P=O bonds [36].”

- Although Scheme 1 is frequently cited, that scheme was not included in the manuscript.

√  Yes, we understand and accept this comment, Scheme 1 had been added.

- Why the maximum temperature used in TGA analysis was 600°C? What happens with the materials at higher temperatures? It seems that the loss of matter had not finished.

√  Yes, we understand and respect this comment,

   Actually, we set the heating temperature of NETZSH TG 209C (TGA instrument) at 600 oC more than 800 oC or higher, because we consider that most or all organic species had been heated to gas under 600 oC, which was sufficient to characterize the immobilization situation of synthesized catalyst.

    Therefore, we sincerely hope this arrangement could be considered in revision.

- Include a comparison of the found catalytic activity with other reported systems.

√  Yes, we understand and accept this comment,

Therefore, a new paragraph was added as the last paragraph of Sect. 2.5.3., as follows:

“From another point of view, the yield stemming from the protocol of Table 4 catalyzed by C2 (entry 12, Table 4; yield of 88%) was much higher than that from the same reaction catalyzed by PdCl2(dppf) (yield of 65%) [43]. Furthermore, C2 also showed very close yield compared to the catalytic combination of PdCl2(dppf) with BMImPF6 [45]. These results clearly indicated that ball milling of divalent palladium(II) precursor with silica gel would produce highly active palladium catalyst.”.

Furthermore, ref. 45 and ref. 46 were exchanged.

- Describe the procedure that was followed for activation of the catalyst before reusing it.

√  Yes, we understand and accept this comment,

   Therefore, in the second paragraph of Sect. 3.5.1.,

   “After cooling to room temperature, the mixture was filtrated under reduced pressure, and the residual solids were collected and recycled after consumables were added.”

   was changed to

   “After cooling to room temperature, the mixture was filtrated under reduced pressure, and the residues (catalyst) were collected, dried in air, and then recycled after consumables (substrates, solvent, base) were added.”.

- Because the presence of water in THE solvent system, were two phases present in the system?

√  Yes, we understand and accept this comment,

   THE solvent system included toluene, water and ethanol. Although toluene is immiscible with water, ethanol is soluble with both toluene and water. In practice, the catalytic solution appeared to be one phase system (suspension containing particles of C0‒C3). This suspension can be separated by filtration under reduced pressure (using Büchner funnel).

- What is the role in the reaction of each of the components of THE solvent system?

√  Yes, we understand and accept this comment,

   Herein, toluene dissolves organic substrates and homogeneous palladium catalyst, water dissolves base (Na2CO3), while ethanol plays as phase transfer reagent. The combination of these three solvents may facilitate the transformation.

- Line 614. Gove some information about the used purification procedure of the solvents.

√  Yes, we understand and accept this comment,

   In Line 614, the two imidazolium ILs were bought from reagent company, whose application and recycling had been shown in Sect. 3.5.2. However, the recycling details needed to be added as follows:

   In the second paragraph of Sect. 3.5.2.,

   “The left IL phase was then mixed with consumables (substrates, solvent and base) for recycling.” was added.

-As the milling beads were of ZrO2, was identified if some material of the beads were deposited on the final catalyst because of the milling and if was there any catalytic effect?

√  Yes, we understand and accept this comment,

   In experiments, we had carefully checked the XPS survey scan for the synthesized catalysts, and found there were no Zr-containing species (Figures 2a‒2d). Therefore, the contamination effects of ball milling seemed very marginal or even none.

    We are really grateful for Reviewer’s very kind and helpful reminding.

- Line 652. What do the authors mean with “under N2 protection”?

√  Yes, we understand and accept this comment,

   “under N2 protection” meant the sample was heated in nitrogen flow. Therefore, in the fourth paragraph of Sect. 3.2.,

   “Thermogravimetric analysis (TGA) was performed on NETZSH TG 209C, having TASC 414/4 controller under nitrogen protection, with a heating rate of 10 °C/min at 30–600 °C.”

was changed to

    “Thermogravimetric analysis (TGA) was performed on NETZSH TG 209C, having TASC 414/4 controller, and sample was heated under nitrogen protection (N2 flow, 40 mL min-1), along with a heating rate of 10 °C min-1 at 30–600 °C.”.

- Line 665. The word “respectively” is not necessary.

√  Yes, we understand and accept this comment,

   Therefore, in Sect. 3.3., “respectively” was deleted.

- Line 666. What was the used stirring rate?

√  Yes, we understand and accept this comment,

Therefore, in the first paragraph of Sect. 3.4.,

“(300 rpm)” was added.

- Line 725. The “THE solvent system” was described in line 705, it is not necessary to detailed again.

√  Yes, we understand and accept this comment,

   Therefore, in the second paragraph of Sect. 3.5.2.,

   “(toluene, 8 mL; H2O, 4 mL; EtOH, 2 mL)” was deleted.

That’s all for revisions according to Reviewer 2.

We all authors are really grateful for Reviewer 2’s very helpful comments and very kind reminding. Thanks!

Reviewer 3 Report

In this paper, authors prepared  PdO2 Nanoparticles Immobilized over silica gel  immobilized PbO2 catalyst through mechanochemical synthesis and used it for Suzuki–Miyaura cross-coupling reactions. The research is significant. Here, some issues should be solved  as follows.

1. Why does the acidic sites increase after addition of Pb?

2. Directly comparing the average conversion between Table 3 and Table 4 is not scientific.

3. How different ligands affect catalytic activity should be discussed in more depth。

4. In the front context, authors declared that the imployment of compound 4 had good effect on performance. However, in the latter section, when half mass of them was used, the compound 2 is superior to compound 4. This is contradict with each other.

5. For the proposed mechnism, the evidence given was clearly insufficient. At least, some literature should be referred based on the previous researches.  

Author Response

For comments from Reviewer 3

Reviewer 3:

In this paper, authors prepared  PdO2 Nanoparticles Immobilized over silica gel  immobilized PbO2 catalyst through mechanochemical synthesis and used it for Suzuki–Miyaura cross-coupling reactions. The research is significant. Here, some issues should be solved  as follows.

√ First of all, we all authors are really grateful for Reviewer 2’s time, evaluation and work on our submission. Thanks.

  1. Why does the acidic sites increase after addition of Pb?

√  Yes, we understand and accept this comment,

   We consider that the acidic sites would be increase after addition of Pd, mainly because the palladium atom may provide more empty orbitals than silicon.

   Therefore, in the last paragraph of Sect. 2.2.,

   “but the immobilization of Pd components would further increase the acidity,” was changed to “but the immobilization of palladium components would further increase the acidity, mainly because palladium atom may provide more empty orbitals than silicon, ”.

  1. Directly comparing the average conversion between Table 3 and Table 4 is not scientific.

√  Yes, we understand and respect this comment, and we think this is a very kind and helpful reminding to this part of work.

   However, in our opinion, this issue can be improved by reducing the scope of comparison, and the results obtained may be more available and significant.

   Therefore, in the first paragraph of Sect. 2.5.2.,

   “According to the first-round catalytic results summarized in Tables 3‒4, it can be seen that the averaged conversion of entries 3-19 in Table 3 (63%) was lower than that of entries 3-19 in Table 4 (67%), indicating 3-methoxycarbonylphenylboronic acid (compound 2, Table 3) was a less effective nucleophile than its para isomer (compound 4, Table 4) in the present Suzuki–Miyaura cross-coupling.”

    was changed to

    “According to Tables 3‒4, the averaged conversion of entries 3-10 in Table 3 (60%) was lower than that of same entries in Table 4 (62%). As for entries 11-15 in Tables 3‒4, Table 3 (64%) still gave less outputs than Table 4 (69%). Additionally, the averaged conversion of entries 16‒19 in Table 3 (66%) was also lower than that in Table 4 (72%). Therefore, 3-methoxycarbonylphenylboronic acid (compound 2, Table 3) seemed to be a less effective nucleophile than its para isomer (compound 4, Table 4).”.

  1. How different ligands affect catalytic activity should be discussed in more depth.

√  Yes, we understand and accept this comment,

   Therefore, at first,

   In the fourth paragraph of Sect. 2.1.,

   “Herein, Pd2+ coming from various divalent palladium precursors was hydrolyzed into Pd(OH)2 under ball milling, which was further oxidized and dehydrated into PdO2 under drying (Scheme 1).” was added.

  1. In the front context, authors declared that the imployment of compound 4 had good effect on performance. However, in the latter section, when half mass of them was used, the compound 2 is superior to compound 4. This is contradict with each other.

√  Yes, we understand and respect this comment,

   After careful check entry 5 of Table 3 and entry 5 of Table 4, the former yield is 93%, while the latter is 92%, these are experimental results and very close, showing the effects of catalyst C1. Overall, the results obtained clearly indicated that C1 (derived from ball milling) had great catalytic activity.

  1. For the proposed mechnism, the evidence given was clearly insufficient. At least, some literature should be referred based on the previous researches.

√  Yes, we understand and accept this comment,

   Therefore, in Sect. 2.6.,

  • In the first paragraph,

“During catalytic process, PdO2 nanoparticles on surface of C1 are first activated and reduced to Pd(0) species by internal electron transfer or residual Pd(OAc)2 incorporated in C1, leading to catalytically active TS1.”

was changed to

“During catalytic process, PdO2 nanoparticles fixed on surface of C1 are first activated and reduced to Pd(0) species by internal electron transfer of C1, leading to catalytically active TS1 [13].”.

  • In the first paragraph,

“Next, the organic iodide (electrophile, compound 1) is coordinated to TS1, and meanwhile surficial PdO2 nanoparticles are oxidized to Pd(II) species under the present conditions, particularly regarding the oxidation of O2 in air.”

was changed to

“Next, the organic iodide (electrophile, compound 1) is coordinated to TS1, and meanwhile surficial PdO2 nanoparticles are oxidized to Pd(II) species under the present conditions, like oxidation of O2 in air [13].”.

That’s all for revisions according to Reviewer 3.

We all authors are really grateful for Reviewer 3’s very helpful comments and very kind reminding. Thanks!

At the end of this revision, the manuscript was further refined and polished, mainly focusing on language improvement, but not changing any data and judgments, as follows:

  • In Abstract,

“In this work, a series of silica gel-supported PdO2 nanoparticles were prepared by ball milling of silica gel” was changed to “In this work, a series of silica gel-supported PdO2 nanoparticles having size of 25‒66 nm were prepared by ball milling of silica gel”.

  • In Abstract,

“This mechanochemical method created PdO2 nanoparticles with size of 25‒66 nm over silica gel.” was deleted.

  • In Abstract,

“This work provided an effective and inexpensive catalyst” was changed to “This work provided a highly active and inexpensive catalyst”.

  • In the first paragraph of Introduction,

“containing” was changed to “including”.

  • In the third paragraph of Introduction,

“That is to say,” was deleted.

  • In the fourth paragraph of Introduction,

“in the presence of” was changed to “with”.

  • In the fifth paragraph of Introduction,

“C-3 functionalization of indazole” was changed to “this transformation”.

  • In the fifth paragraph of Introduction,

“, and significant results were obtained” was deleted.

  • In the fifth paragraph of Introduction,

“otherwise leading to unwanted by-products” was changed to “otherwise unwanted by-products appeared”.

  • In the sixth paragraph of Introduction,

“a lot of” was changed to “many”.

  • In the tenth paragraph of Introduction,

“of synthesized catalyst” was deleted.

  • In the eleventh paragraph of Introduction,

“Additionally, iron oxide was ever employed as support for immobilizing single-atom palladium components by using ball milling, and the catalyst showed great activity for Suzuki-Miyaura cross-coupling” was changed to “Iron oxide was also employed as support for immobilizing single-atom palladium components through ball milling, and the catalyst showed great activity too”.

  • In the twelfth paragraph of Introduction,

“also” was changed to “ever”.

  • In the first paragraph of Sect. 2.1.,

“Pd” was changed to “palladium”.

  • In the second paragraph of Sect. 2.1.,

“Pd” was changed to “palladium”.

  • In the third paragraph of Sect. 2.1.,

“On the other hand,” was deleted.

  • In the fifth paragraph of Sect. 2.1.,

“on the immobilization of palladium” was changed to “on immobilization”.

  • In the fifth paragraph of Sect. 2.1.,

“These components were derived from the organic ligands of Pd precursors that used in ball milling synthesis of C1‒C3 (Scheme 1).” was changed to “These components may be derived from the organic ligands of palladium precursors (Scheme 1).”.

  • In the sixth paragraph of Sect. 2.1.,

“XRD” was added.

  • In the second paragraph of Sect. 2.2.,

“In view of porosity,” was changed to “The”.

  • In the second paragraph of Sect. 2.3.,

“immobilization” was deleted.

  • In the third paragraph of Sect. 2.3.,

“In order to further study the immobilization details, UV-Vis spectroscopy was employed when C0-C3 were dispersed into water, respectively.” was changed to “UV-Vis spectroscopy was employed to study the immobilization from another point of view.”.

  • In the third paragraph of Sect. 2.3.,

“of organic species appeared in samples” was changed to “on organic species of samples”.

  • In Sect. 2.5.2,

“under various catalytic profiles.” was added.

  • In the third paragraph of Sect. 2.5.3.,

“(palladium precursor of C1, Scheme 1)” was deleted.

  • In the first paragraph of Sect. 2.5.4.,

“()” was added.

  • In Sect. 2.5.5.,

“Temperature also showed important influences on catalytic outputs.” was deleted.

  • In Sect. 2.5.6.,

“It was also necessary to test role of solvent during catalysis, which meant a lot to further optimization of the reaction.” was deleted.

  • In the third paragraph of Sect. 2.5.6.,

“However, as far as was known, the suitable palladium species mainly referred to homogeneous palladium salts and complexes rather than supported palladium materials.”

was changed to

“However, the suitable palladium species may be limited to homogeneous palladium salts or complexes rather than supported palladium materials.”.

  • In the first paragraph of Sect. 2.6.,

“as shown” was deleted.

  • In the first paragraph of Conclusions,

“with phenyl groups” was deleted.

  • In the second paragraph of Conclusions,

“in Suzuki–Miyaura cross-coupling” was deleted.

  • In the last paragraph of Conclusions,

“Obviously, this work provided a highly active and inexpensive protocol for C-3 modification of 1H-indazole.” was added.

That’s all for this revision.

We all authors are really grateful for Editor’s and Reviewers’ very helpful instructions, comments and very kind reminding.

Thanks!

Round 2

Reviewer 1 Report

[The reviewer is mainly concerned with the section of synthetic methods].

The reviewer has just read the revised manuscript.  New Scheme 1 is useful for reader’s comprehension.

In many parts, further editing efforts will be required for the conciseness and clarity.    

On the whole, the reviewer recommends the publication in Molecules after revision, especially, full editing works.

<Minor comments and suggestions>

1.     line 404; “Overall” should be deleted.

2.     line 452; “Furthermore” should be deleted.

Author Response

The Second Time Responses to Referees’ Comments

A List of Changes

For comments from Reviewer 1

Reviewer 1:

The reviewer has just read the revised manuscript.  New Scheme 1 is useful for reader’s comprehension.

√  First of all, we all authors are really grateful for Reviewer 1’s time, evaluation and work on our submission for the second time.

We had added Scheme 1 that missed in the submission of last time.

Thanks.

In many parts, further editing efforts will be required for the conciseness and clarity.

√ Yes, we understand and accept this comment,

And therefore,

  • In Abstract,

“The C-3 modification of 1H-indazole produced active pharmaceuticals for treatment of illnesses such as cancer and HIV.” was changed to “The C-3 modification of 1H-indazole produced active pharmaceuticals for treatment of cancer and HIV.”.

  • In Abstract,

     “due to lack of powerful method for C-C bond formation on less reactive C-3 position.” was changed to “due to lack of efficient C-C bond formation on less reactive C-3 position.”.

  • In Abstract,

  “in water” was deleted.

  • In Abstract,

  “but that of synthesized catalysts became” was changed to “the synthesized catalysts showed”.

  • In Abstract,

“This work provided a highly active and inexpensive catalyst for C-3 modification of 1H-indazole, which meant a lot to the large-scale production of very useful 1H-indazole-based pharmaceuticals.” was changed to “This work provided a highly active and inexpensive series of catalysts for C-3 modification of 1H-indazole, which meant a lot to the large-scale production of 1H-indazole-based pharmaceuticals.”.

  • In the first paragraph of Introduction,

“Indazoles referred to a group of nitrogen-containing bicyclic compounds, including an electron-rich pyrazole and a fused benzene ring [1]. Indazoles can be also regarded as a nitrogen-substituted product of indole, but their studies seemed much less than those of indole for a long time [1].” was changed to

  “Indazoles referred to a group of bicyclic compounds containing an electron-rich pyrazole and a fused benzene ring [1], which can be regarded as a nitrogen-substituted product of indole, but their studies seemed much less than indole for a long time [1].”

  • In the third paragraph of Introduction,

  “Nowadays, the attempts on synthesis or functionalization of 1H-indazole had attracted continuous attentions from the fields of organic synthesis and pharmaceutical discovery.” was changed to

  “Nowadays, the attempts on synthesis and functionalization of 1H-indazole had attracted wide and continuous attentions.”

  • In the third paragraph of Introduction,

“However, these profiles usually suffered from a big problem in common. The lack of substrate availability and subsequent product diversity actually depressed large-scale production of versatile 1H-indazole-based pharmaceuticals (Figure 1) [8‒10].” was changed to

“However, these profiles were usually lack of substrate and product diversity, obviously depressing large-scale production of versatile 1H-indazole-based pharmaceuticals (Figure 1) [8‒10].”.

  • In the forth paragraph of Introduction,

  “In the past two decades,” was deleted.

  • In the forth paragraph of Introduction,

“of ” was changed to “on”.

  • In the fifth paragraph of Introduction,

“demanded a considerably irreversible protection of N-H group of 1H-indazole,” was changed to “demanded a tight (but irreversible sometimes) protection of N-H group on 1H-indazole,”.

  • In the fifth paragraph of Introduction,

“only needed a highly removable” was changed to “just needed an easily removable”.

  • In the sixth paragraph of Introduction,

“as well as the lack of profile for catalyst recovery [13,14]” was changed to “as well as the lack of effective catalyst recovery [13,14]”.

  • In the sixth paragraph of Introduction,

“Therefore, many attempts on immobilization of palladium into various supports were carried out in order for simplifying catalyst synthesis and improving catalyst recovery.” was changed to “Therefore, the immobilization of palladium into support was carried out in order for simplifying catalyst synthesis and improving catalyst recovery.”.

  • In the second paragraph of Sect. 2.1.,

“Pd” was changed to “palladium”.

  • In the third paragraph of Sect. 2.1.,

“Pd” was changed to “palladium”.

  • In the forth paragraph of Sect. 2.1.,

“Pd” was changed to “palladium”.

  • In the sixth paragraph of Sect. 2.1.,

“which could be ascribed to” was changed to “corresponding to”.

  • In the first paragraph of Sect. 2.2.,

“With the composition and crystalline data obtained so far, it was necessary to further test the textural and other physicochemical properties of synthesized samples in order for better understanding their difference.” was changed to “In order to further understand difference of catalysts, the textural and other physicochemical properties of synthesized samples were tested.”.

  • In the second paragraph of Sect. 2.2.,

“component” and “components” were deleted.

  • In the third paragraph of Sect. 2.3.,

“which can be ascribed to” was changed to “all corresponding to”.

  • In the third paragraph of Sect. 2.3.,

“at 325 cm or higher” was changed to “above 325 nm”.

  • In the third paragraph of Sect. 2.3.,

“following” was deleted.

  • In the third paragraph of Sect. 2.3.,

“already” was deleted.

  • In the forth paragraph of Sect. 2.3.,

“It was also interesting to study the chemical compositions of synthesized catalysts quantitatively by using thermogravimetric analysis (TGA).” was changed to “It was also interesting to study composition of synthesized catalyst by using through TGA.”.

  • In the forth paragraph of Sect. 2.3.,

“First of all,” was changed to “The”.

  • In the forth paragraph of Sect. 2.3.,

  “from silica gel under heating” was deleted.

  • In the forth paragraph of Sect. 2.3.,

“which can be attributed to” was changed to “meaning”.

  • In the fifth paragraph of Sect. 2.3.,

“through ball milling” was deleted.

  • In the sixth paragraph of Sect. 2.3.,

“that of” was deleted.

  • In the first paragraph of Sect. 2.4.,

“The abovementioned physicochemical properties could be further illustrated by studying their morphologies and internal structures. Above all,” was changed to “The”.

  • In the first paragraph of Sect. 2.4.,

“furthermore,” was deleted.

  • In the second paragraph of Sect. 2.4.,

“After immobilization of Pd(OAc)2 under ball milling, the resulting product (C1)” was changed to “The C1”.

  • In the forth paragraph of Sect. 2.4.,

“play” was changed to “played”.

  • In the fifth paragraph of Sect. 2.4.,

“and illustrate” was deleted.

  • In the first paragraph of Sect. 2.5.2.,

“First of all” and “Secondly,” were deleted.

  • In the third paragraph of Sect. 2.5.3.,

“it can be seen that” was deleted.

“Pd” was changed to “palladium”.

  • In the fifth paragraph of Sect. 2.5.3.,

“the order of activity found among synthesized catalysts” was changed to “the activity order of synthesized catalysts”.

  • In the first paragraph of Sect. 2.5.4,

“The optimization of reaction parameters such as loadings of catalyst and base was beneficial to amplification of reaction.” was deleted.

  • In the first paragraph of Sect. 2.5.6.,

“Above all,” was changed to “However,”.

  • In the first paragraph of Sect. 2.5.6.,

“species” was changed to “components”.

  • In the last paragraph of Sect. 2.5.7.,

“a much less effective attempt” was changed to “not effective”.

  • In the first paragraph of Sect. 2.6.,

“In order to further understand C-3 modification of 1H-indazole over PdO2 nanoparticles,” was deleted.

“During catalytic process,” was changed to “The”.

  • In the second paragraph of Sect. 3.1.,

“were bought from” was changed to “were commercially available from”.

  • In the first paragraph of Conclusions,

“those of silica gel” was changed to “silica gel backbone”.

  • In the second paragraph of Conclusions,

  “palladium” was deleted.

  • In the second paragraph of Conclusions,

  “palladium” was deleted.

  • In the last paragraph of Conclusions,

  “Obviously, this work provided a highly active and inexpensive protocol for C-3 modification of 1H-indazole.” was changed to “Obviously, this work would contribute to the exploration of 1H-indazole-based pharmaceuticals.”.

On the whole, the reviewer recommends the publication in Molecules after revision, especially, full editing works.

√ Yes, we understand and accept this comment, and meanwhile we all authors are really grateful for Reviewer 1’s recommendation.

<Minor comments and suggestions>

  1. line 404; “Overall” should be deleted.

√ Yes, we understand and accept this comment,

   Therefore, in line 404 (first paragraph of Sect. 2.5.3),

   “Overall, ” was deleted.

  1. line 452; “Furthermore” should be deleted.

√ Yes, we understand and accept this comment,

Therefore, in line 452 (first paragraph of Sect. 2.5.3),

“Furthermore,” was deleted.

That’s all for revisions according to Reviewer 1.

We all authors are really grateful for Reviewer 1’s very helpful comments and recommendation.

Thanks.

That’s all for this revision (the second time).

We all authors are really grateful for Editor’s and Reviewer 1’s very helpful instructions, comments and recommendation.

Thanks.
